# A SHERLOCK toolbox for eco-epidemiological surveillance of African trypanosomes in domestic pigs from Western Africa

Roger Eloiflin[1†], Elena Pérez-Antón[2†], Aïssata Camara[3†], Annick Dujeancourt-Henry[2], Salimatou Boiro[3], Martial N Djetchi[4], Mélika Barkissa Traoré[4], Mathurin Koffi[4], Dramane Kaba[5], Yann Le Pennec[6], Bakary Doukouré[6], Abdoulaye Dansy Camara[7], Moïse Kagbadouno[7], Pascal Campagne[8], Mamadou Camara[7], Vincent Jamonneau[1], Sophie Thévenon[1,9], Jean-Mathieu Bart[1,7], Lucy Glover[2*], Brice Rotureau[3,10*]

[1]INTERTRYP, Université de Montpellier, Cirad, IRD, Montpellier, France; [2]Trypanosome Molecular Biology Unit, Institut Pasteur, Université de Paris, Paris, France; [3]Parasitology Unit, Institut Pasteur of Guinea, Conakry, Guinea; [4]Unité de Formation et de Recherche Environnement, Université Jean Lorougnon Guédé, Daloa, Côte d'Ivoire; [5]Unité de Recherche Trypanosomoses, Institut Pierre Richet, Bouaké, Côte d'Ivoire; [6]Virology Unit, Institut Pasteur of Guinea, Conakry, Guinea; [7]Programme National de Lutte contre la Trypanosomiase Humaine Africaine, Ministère de la Santé, Conakry, Guinea; [8]Bioinformatics and biostatistics hub, Institut Pasteur, Paris, France; [9]Cirad, UMR INTERTRYP, Montpellier, France; [10]Trypanosome Transmission Group, Trypanosome Cell Biology Unit, Institut Pasteur, Université Paris Cité, Paris, France

*For correspondence:
lucy.glover@pasteur.fr (LG);
rotureau@pasteur.fr (BR)

†These authors contributed equally to this work

Competing interest: The authors declare that no competing interests exist.

## eLife Assessment

This **important** study reports an advancement in the diagnosis of Animal African Trypanosomosis (AAT), which adapts a CRISPR-based diagnostic tool (SHERLOCK4AAT) to detect different trypanosome species responsible for AAT. The evidence supporting the conclusions is **convincing** and in line with the current state-of-the-art diagnostics. This study will be of interest to the fields of Epidemiology, Public Health, and Veterinary Medicine.

**Abstract** Animal African trypanosomosis (AAT), caused by protist parasites of the genus *Trypanosoma*, puts upward of a million head of livestock at risk across 37 countries in Africa. The economic impact of AAT and the presence of human-infectious trypanosomes in animals place a clear importance on improving diagnostics for animal trypanosomes to map the distribution of the veterinary parasites and identify reservoirs of human-infectious trypanosomes. We have adapted the CRISPR-based detection toolkit SHERLOCK (Specific High-sensitivity Enzymatic Reporter unLOCKing) for trypanosomatid parasites responsible for AAT (SHERLOCK4AAT) including Pan-trypanosomatid, *Trypanozoon*, *T. vivax*, *T. congolense*, *T. theileri*, *T. simiae*, and *T. suis* assays. To test the applicability of this technique in the field, we analysed dried blood spots collected from 200 farm and 224 free-ranging pigs in endemic and historical human African trypanosomiasis foci in Guinea and Côte d'Ivoire, respectively. The results revealed that SHERLOCK4AAT can detect and discriminate between trypanosome species involved in multiple infections with a high sensitivity. 62.7% [58.1,

67.3] of pigs were found infected with at least one trypanosome species. *T. brucei gambiense*, a human-infectious trypanosome, was found in one animal at both sites, highlighting the risk that these animals may act as persistent reservoirs. These data suggest that, due to their proximity to humans and their attractiveness to tsetse flies, pigs could act as sentinels to monitor *T. b. gambiense* circulation using the SHERLOCK4AAT toolbox.

## Introduction

Through a combination of vector control, reactive screening, passive diagnosis, new therapeutic drugs and the support of national control programmes, donors and stakeholders, interrupting the transmission of gambiense human African trypanosomiasis (HAT) at pan-African level is in reach (*Barrett et al., 2024*). Reported annual cases have been maintained below 1000 since 2018, making the elimination (zero transmission) of gambiense HAT (gHAT) by 2030, one of the major objectives of the World Health Organisation (WHO), achievable (*Franco et al., 2024*). Nevertheless, animal African trypanosomosis (AAT), also known as *nagana*, remains a significant burden to sustainable agricultural development, and represents a threat to livestock and smallholder economies (*Boulangé et al., 2022*), with a total estimated loss of 4.75 billion USD per year in terms of agricultural gross domestic product in tsetse-infested lands (*Budd, 2001*). Unlike HAT, which is caused by either *Trypanosoma brucei (T. b.) gambiense* in West Africa, or *T. b. rhodesiense* in East Africa, AAT encompasses a wider range of trypanosomatid parasites. In sub-Saharan Africa, animal-infecting trypanosomes including *T. vivax*, *T. congolense*, *T. brucei*, *T. theileri*, *T. simiae*, and *T. suis* are transmitted by blood-sucking flies, such as Glossinidae and Tabanidae, to a wide range of wild and domestic mammals. Socio-environmental factors such as land cover, humidity, temperature, and livestock migration impact trypanosome transmission (*Karshima et al., 2016*). This is particularly evident in rural areas of sub-Saharan Africa, where high incidences of *T. brucei*, *T. congolense*, and *T. vivax* are observed, due to conditions favourable to the proliferation of flies (*Okello et al., 2022b*).

Symptoms of AAT are host and pathogen-dependent, with an infection potentially leading to spontaneous abortions, reduction in milk and meat production, delays in sexual maturity, low calving rates, loss of draught power, and ultimately to high mortality, all of which compound poverty in many sub-Saharan countries (*Kristjanson et al., 1999*). Indirect control strategies, including the combination of vector control, breeding of trypanotolerant animals, and avoiding grazing in infested areas, have proved to be efficient in reducing the number of infections (*Meyer et al., 2016*). Chemotherapy and chemoprophylaxis are the main means for controlling AAT, although inadequate usage, presence of fake drugs and emergent drug resistance are hampering trypanocide efficacy (*Giordani et al., 2016*; *Richards et al., 2021*). Reliable and adapted diagnostic tests are crucial for parasite detection to better target infected animals for treatment, as well as for parasite identification to understand the distribution of trypanosomes in endemic areas. Microscopic, serological and molecular methods are generally used to diagnose AAT. Microscopy-based methods are the most commonly used diagnostic methods, but lack sensitivity with a limit of detection (LoD) ranging from $10^4$ to $10^5$ parasites/ml when using fresh blood, to 250–5000 parasites/ml for methods combining centrifugation (*Desquesnes et al., 2022a*). Serological methods, such as the card agglutination test (CATT) and ELISA, can be used to identify anti-*Trypanosoma* antibodies, but their reliability is often compromised by the poor antigen characterization in animal-infecting trypanosomes or by the detection of antibodies from previous infections. In contrast, molecular diagnostic techniques, including PCR and Loop-mediated isothermal amplification (LAMP), offer increased sensitivity (73–100%) and LoD (1–1000 parasites/ml) by targeting specific regions of the parasites' DNA. For the main AAT-related *Trypanosoma* species, pan-species primers (targeting ITS1 for instance) and genus/species/type-specific primers are available and currently used to detect the DNA of parasites from the subgenus *Nannomonas* (*T. congolense*), *Duttonella* (*T. vivax*), and *Trypanozoon* (*T.b.brucei*, *T. evansi*, and *T. equiperdum*) (*Álvarez-Rodríguez et al., 2022*; *Desquesnes et al., 2022a*).

AAT-related trypanosome species circulate co-endemically with *T. b. gambiense* in West and Central Africa, where domestic animals such as pigs, dogs, cattle, sheep, and goats are suspected reservoirs (*Magang et al., 2023*; *Traoré et al., 2021*). In these regions, Suids (warthogs, domestic, and bush pigs) and bovids (bushbucks and buffalos) are the preferred hosts for most tsetse fly species (*Serem et al., 2024*). In a historical HAT focus in Côte d'Ivoire, free-range pigs sharing the same environment

as humans were identified as reservoirs for *T. b. gambiense* (*Traoré et al., 2021*). The diagnostic techniques employed in this study, involving single round (*Radwanska et al., 2002*) or nested PCR (*Morrison et al., 2008*), however, tended to overestimate the prevalence of the human infective parasites in pigs (*Traoré et al., 2021*). Therefore, an active and accurate surveillance of the infection status of domestic animals using a more reliable diagnostic tool will be key to reach and sustain gHAT elimination (*Simo and Rayaisse, 2015*; *Vourchakbé et al., 2020*). In addition, a comprehensive map of the animal infective trypanosomes circulating in tsetse-infested areas would enable an improved control strategy and prevent or limit further economic losses.

In this context, developing new diagnostic tools that are species-specific and highly sensitive is essential to accurately diagnose infected individuals and define the circulation of trypanosomes in domestic animals. For this purpose, we have adapted the Specific High-sensitivity Enzymatic Reporter unLOCKing (SHERLOCK) detection that offers unprecedented levels of diagnostic specificity and sensitivity (*Gootenberg et al., 2017*), and has been adapted for the detection of HAT (*Sima et al., 2022*). SHERLOCK combines an isothermal amplification by recombinase polymerase (RPA) with Cas13a detection of the target RNA (*Gootenberg et al., 2017*; *Kellner et al., 2019*; *Piepenburg et al., 2006*). The RPA step, in combination with a reverse transcriptase, allows the use of total nucleic acid (TNA) as an input, increasing the flexibility of the assay. Here, we have developed a broad panel of SHERLOCK assays for the detection of the six most prevalent AAT-related species. We have applied the SHER-LOCK4AAT panel to two populations of domestic pigs from two eco-epidemiological regions of West Africa to determine their importance as reservoirs of both animal and human-infective trypanosomes. A total of 424 blood samples were collected (1) from free-ranging pigs in the last hypo-endemic foci of gHAT in Côte d'Ivoire, and (2) from farm pigs bred in the historical gHAT focus of forested Guinea where unconfirmed seropositive cases of gHAT are still sporadically detected (*Courtin et al., 2019*). These samples were screened using the complete SHERLOCK4AAT panel to provide an in-depth profile of the trypanosome species circulating among these domestic animals.

## Results

### Selection of targets for broad and species-specific SHERLOCK assays targeting AAT species (SHERLOCK4AAT)

Target gene selection was based on two requirements: to develop (1) a broad SHERLOCK assay for the detection of all trypanosomatid parasites and (2) for species discrimination. To adapt SHER-LOCK for the detection of AAT-related species, we followed the criteria as described in *Sima et al., 2022*. Briefly, candidate genes must (1) be expressed in the bloodstream form of the parasites, (2) be conserved between strains, (3) have few or no single-nucleotide polymorphisms (SNPs), or (4) show sequence divergence that would allow for species discrimination. All RPA primers and crRNAs designed for each SHERLOCK assay are listed in *Supplementary file 2*.

#### Pan-trypanosomatid assay

SHERLOCK has the capacity to discriminate between two sequences differing by a single nucleotide (*Gootenberg et al., 2017*). This study therefore focused on regions of the *18S rRNA* gene that show limited sequence homology between species. In addition to being a multicopy gene, the *18S rRNA* gene shares over 70% identity between Kinetoplastea members (*Figure 1—figure supplement 1*). Using BLAST analysis, the region between 1535–1672 bp (XR_002989632.1) was found to be conserved between all trypanosomatid species and was selected as the pan-trypanosomatid SHER-LOCK target. RPA primers were designed to amplify this 138 bp region using Primer-Blast (*Ye et al., 2012*; *Figure 1—figure supplement 2A* and *Supplementary file 2*). The level of identity of the target region was assessed between the different sequences described in the Trypanosomatidae family, as well as in the order, subclass, and class that contain this family (*Supplementary file 4*). Examination of the Euglenozoa phylum, excluding the Kinetoplastea class, showed 98.4% identity within the Trypano-somatidae family and 97.3% identity within the Kinetoplastea class (*Supplementary file 4*). This was reduced to 82.9% for the Phylum Euglenozoa, excluding the Kinetoplastea sequences (*Supplementary file 4*). Three distinct crRNAs were designed to this region, which can be used individually or multiplexed to increase the sensitivity (*Figure 1—figure supplement 3A* and *Supplementary file 2*). The sequences described in each group were 100% conserved to the crRNA. RPA primers and

crRNA targets of the pan-trypanosomatid SHERLOCK assay target regions (Kinetoplastea) were then compared to the homologous sequences from other classes within the Phylum Euglenozoa (Euglenida, Diplonemea, Symbiontida, and Kinetoplastea) (*Figure 1—figure supplement 2A* and *Supplementary file 4*). In addition, sequences from related protozoan parasite classes (Diplomonadida, Haemosporida, and Ciliophora) and from the genetically close but phylogenetically distinct group Microsporidia were also included in the analysis (*Figure 1—figure supplement 2A*).

## Pan-*Trypanozoon* assay

A pan-*Trypanozoon* SHERLOCK assay was designed for increased selectivity to especially detect *T. brucei s.l.* (sensu lato). The regions within the 18S rRNA that were highly conserved among the species of the *Trypanozoon* subgenus, while showing significant divergence from other trypanosome species not belonging to the *Trypanozoon* subgenus, were analysed (*Figure 1—figure supplement 2B*). A region ranging from 1318 to 1433 bp (XR_002989632.1) (*Figure 1A* and *Supplementary file 2*) was selected as it showed high conservation among the *Trypanozoon* subgenus (99.12–100% percentage of identity, query coverage >90%), except for three *T. evansi* sequences (91.96–96.49%, OL869590.1, KT844944.1, MF142286.1). The crRNA target region selected was also conserved in the subgenus *Trypanozoon* and divergent in other parasites. Specific alignments with trypanosome species that could share the same hosts (*T. congolense*, *T. vivax*, *Leishmania* spp., etc) were performed to assess target amplicon specificity (*Figure 1—figure supplement 2B*). The pan-*Trypanozoon* SHERLOCK assay can discriminate between the species belonging to the *Trypanozoon* subgenus (*T. b.* spp., *T. evansi* types A and B, and *T. equiperdum*) from other non-*Trypanozoon* trypanosomes or other co-endemic parasites.

## Species-specific assays

For species discrimination within a single sample, several SHERLOCK assays were developed to detect the most prevalent species causing AAT. These include *T. congolense*, *T. vivax*, *T. theileri*, *T. simiae*, and *T. suis*.

### T. congolense

Regions conserved in the *18S rRNA* gene for all described *T. congolense* sequences, including the Savannah, Forest, Kilifi, and Dzanga-Sangha groups, were analysed. The target region selected from position 116 to 246 bp of the *18S rRNA* gene (AJ009146.1) shows 100% identity between the described *T. congolense* sequences. BLAST analysis of the *T. congolense 18S rRNA* gene sequences, excluding the target species (*T. congolense* (taxid: 5692)) was also performed to determine the percentage of identity of closely related species. *T. simiae* (86.72–90.62%), *T. sapaensis* (88.28%), *T. godfreyi* (88.28%), and *T. otospermophili* (87.02%) had the highest percentage identity observed (*Figure 1—figure supplement 4*). Species with common hosts and of veterinary relevance such as the two analysed *T. vivax* strains showed 82.81% and 87.5% identity. In addition, none of these sequences were fully conserved for the selected crRNA (*Figure 1—figure supplement 4*), ensuring the specificity of the test and the absence of cross-detection with other species.

### T. vivax

The *invariant flagellum antigen* (*IFX*) gene has no apparent homologue in any other *Trypanosoma* species or other related parasites, and was therefore selected (*Autheman et al., 2021*; *Romero-Ramirez et al., 2022*). BLAST analysis, using *T. vivax* Y486 (HE573024.1) as a reference genome, resulted in no matches (*Figure 1A* and *Supplementary file 2*).

### T. theileri

A conserved region between the three *T. theileri* lineages (TthI, TthII, and TthIII) (*Brotánková et al., 2022*) was selected. The target region, ranging from 1014 to 1158 bp of the *18S rRNA* gene (AJ009163.1), has 100% identity between the described sequences of the *T. theileri* lineages, and the most closely related sequences were from *T. cruzi*, *T. rangeli*, and *T. dionisii*, with a 97.89% identity (query coverage = 65%). The segment 1052–1081 bp corresponding to the crRNA sequence is highly conserved between *T. theileri* variants (*Figure 1A* and *Supplementary file 2*); however, it contains an SNP (1060 C/T) in a number of the sequences analysed. Hence, one RPA primer pair and two

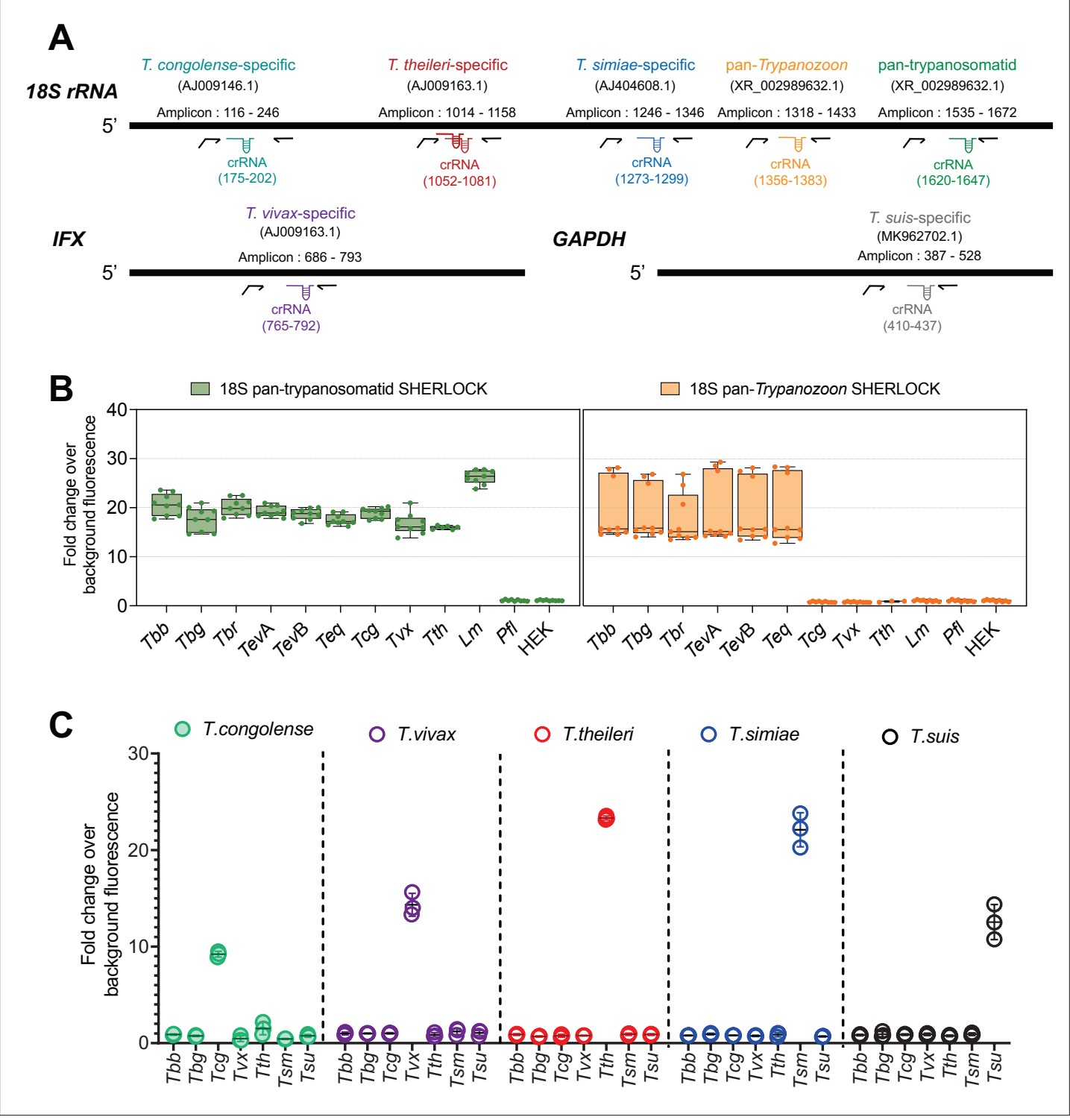

**Figure 1.** Pan and species-specific SHERLOCK assays for human and animal African trypanosome infections. (**A**) Schematic of the target regions used in the SHERLOCK4AAT toolbox. *T. congolense*, *T. theileri* (multiplex), *T. simiae*, Pan-*Trypanozoo*, and Pan-trypanosomatid SHERLOCK assays were designed using the *18S rRNA* gene as a target; *T. vivax* SHERLOCK assay using the *IFX* gene and *T. suis* test using the *gGAPDH* gene. Gene-specific accession number in brackets below the assay name. Details of the regions amplified are provided for each SHERLOCK assay (amplicon). RPA primers are represented by black arrows, the extension in the forward primer represents the T7 polymerase promoter sequence. The regions targeted by the crRNAs (CRISPR guides) are also indicated in the diagram. The schematic representation is to scale. (**B**) Evaluation of the specificity of single pan-trypanosomatid and pan-*Trypanozoon* SHERLOCK tests RNA from *T. b. brucei* (*Tbb*), *T. b. gambiense* (*Tbg*), *T. b. rhodesiense* (*Tbr*), *T. evansi* A (*TevA*)

*Figure 1 continued on next page*

*Figure 1 continued*

and B (*TevB*), *T. equiperdum* (*Teq*), *T. congolense* (*Tcg*), *T. vivax* (*Tvx*), *T. theileri* (*Tth*), *Leishmania major* (*Lm*), *Plasmodium falciparum* (*Pfl*), and human embryonic kidney (HEK) cells. The boxplot means are represented by single lines, and error bars correspond to minimum and maximum values. *N* = 3 for the RPA amplifications and three Cas-13a detections for each RPA amplification, totalling nine replicates for each assay. *T. theileri* was used in fewer replicates as an input due to the limited genetic material available. (**C**) Evaluation of the specificity of *T. congolense*, *T. vivax*, *T. theileri*, *T. simiae*, and *T. suis*-specific SHERLOCK tests on RNA from *Tbb*, *Tbg*, *Tcg*, *Tvx*, *Tth*. Synthetic controls for *T. simiae* (*Tsm*) and *T. suis* (*Tsu*) were used to evaluate the species specificity of the assays. *N* = 1 RPA amplification and three Cas-13a detections totalling three replicates for each assay. Error bars represent standard deviations from the mean of the replicates. All inputs were used at a concentration of 5 ng/µl.

The online version of this article includes the following source data and figure supplement(s) for figure 1:

**Source data 1.** Evaluation of the specificity of single pan-trypanosomatid and pan-Trypanozoon SHERLOCK tests.

**Source data 2.** Evaluation of the specificity of *T. congolense*, *T. vivax*, *T. theileri*, *T. simiae* and *T. suis*-specific SHERLOCK tests.

**Figure supplement 1.** Representation of the proportions of identity between *18S rRNA* and *GAPDH* genes from Microsporidia, Ciliophora, Haemosporida, Diplomonadida, Symbiontida, Diplonemea, Euglenida, and Kinetoplastea, focusing on several species within the last family.

**Figure supplement 2.** Alignments of 18S Pan-trypanosomatid and 18S Pan-Trypanozoon SHERLOCK target regions.

**Figure supplement 3.** Screening of different RPA pairs of primers and/or crRNA guides for each SHERLOCK assay optimization.

**Figure supplement 3—source data 1.** CRISPR guides screening for single 18S pan-trypanosomatid SHERLOCK (A), pan-Trypanozoon SHERLOCK (B), IFX T. vivax-specific SHERLOCK (C), 18S *T. congolense*-specific SHERLOCK (D) and 18S *T. theileri*-specific SHERLOCK (E) assays.

**Figure supplement 4.** Alignments of the 18S *T. congolense* SHERLOCK target region.

multiplexed crRNAs were designed to target both versions of the SNP for the *T. theileri* SHERLOCK assay (*Figure 1—figure supplement 3E*).

### T. simiae

The region from 1246 to 1346 bp of the *18S rRNA* gene (AJ404608.1) was selected as the targeted region for RPA amplification of the *T. simiae*-specific SHERLOCK assay. The segment 1273–1299 bp that corresponds to the crRNA sequence has 100% identity between the described sequences of *T. simiae* (*Figure 1A* and *Supplementary file 2*).

### T. suis

The multicopy *GAPDH* gene was used, as it can be phylogenetically separated into *T. suis* and *T. suis*-like strains from other African trypanosomes (*Rodrigues et al., 2020*). The region from 389 to 528 bp of the *GAPDH* gene (MK962702.1) has been selected as the targeted region for RPA amplification of the *T. suis*-specific SHERLOCK assay (*Figure 1A* and *Supplementary file 2*). This region is 100% conserved between the *T. suis* species described, and outside this clade, the most closely related sequences were found in *T. congolense*, with 90% of identity (query coverage >90%). The segment 410–437 bp of the *gGAPDH* gene (MK962702.1), that corresponds to the crRNA target region, is highly conserved between *T. suis* species, with 100% identity, and the BLAST analysis showed that this target fragment can only align to *T. suis* sequences but not to that of other Kinetoplastea species (taxid:5653).

## SHERLOCK can discriminate between closely related AAT-causing trypanosomatid species

All RPA primers and crRNAs for each targeted species were screened to select the optimal combinations, with the highest specificity and selectivity. For the pan-trypanosomatid assay, RPA primer pair A was used to screen three crRNAs (A:crRNA1, A:crRNA2, and A:crRNA3; *Figure 1—figure supplement 3A*). From this, crRNA 3 was first selected for the simple pan-trypanosomatid assay. For the pan-*Trypanozoon* assay, three RPA primer pairs (A, B, and C) were screened with two crRNA candidates each (A:crRNA1, A:crRNA2, B:crRNA1, B:crRNA2, C:crRNA1, and C:crRNA2, *Figure 1—figure supplement 3B*). The A:crRNA1 combination was selected for the pan-*Trypanozoon* assay, being the one that obtained a higher fold change for the positive control and a lower fold change for the negative control. For the IFX *T. vivax* species-specific SHERLOCK assay, two RPA primer pairs were assessed (A and B) in combination with six crRNA candidates (A:crRNA1, A:crRNA2, A:crRNA3, B:crRNA4, B:crRNA5, and B:crRNA6). RPA primer pair B and crRNA 6 showed the lowest variability between replicates and without any cross-reaction with other trypanosome species (*Figure 1—figure*

supplement 3C). For the *T. congolense*-specific assay, one pair of RPA primer (A) was combined with ten crRNA candidates (A:crRNA1, A:crRNA2, A:crRNA3, A:crRNA4, A:crRNA5, A:crRNA6, A:crRNA7, A:crRNA8, A:crRNA9, and A:crRNA10). The crRNA1 that exhibited the lowest inter-replicate deviation and no cross-reactivity when exposed to the genetic material of non-target trypanosomatid parasites was selected (*Figure 1—figure supplement 3D*). For the *T. theileri*-specific SHERLOCK assay, the combination of the RPA primer pair A with the two designed crRNAs (1 and 2) was used to cover the specific detection of all the *T. theileri* described sequences (*Figure 1—figure supplement 3E*). For *T. simiae* and *T. suis*, single combinations of RPA primer and crRNA were tested. The specificity of each of these combinations was also verified by comparing the specific recognition of each RNA target to RNA from human embryonic kidney (HEK) cell (for pan-trypanosomatid and pan-*Trypanozoon* tests) or RNA from *T. brucei* (for *T. vivax*, *T. congolense*, and *T. theileri* tests) (*Figure 1—figure supplement 3*). In total, one RPA primer–crRNA combination was selected for each assay, based on the highest fold-change over background fluorescence value, the highest specificity, and the minimum standard deviation (*Figure 1A*).

The specificity of the RPA primer–crRNA combinations selected for each assay was then cross-assessed against closely related parasites (*Figure 1B, C*). The single pan-trypanosomatid assay detected RNA from all trypanosomatid species tested (*T. b. brucei*, *T. b. gambiense*, *T. b. rhodesiense*, *T. evansi* (type A and B), *T. equiperdum*, *T. congolense*, *T. vivax*, *T. theileri*, *T. cruzi*, and *Leishmania major*) but not from HEK cells, synthetic controls for *T. simiae* and *T. suis* or co-endemic parasites such as *Plasmodium falciparum* (*Figure 1B*). The pan-*Trypanozoon* assay detected only RNA from members of the *Trypanozoon* subgenus (*T. b. brucei, T. b. gambiense, T. b. rhodesiense, T. evansi* (type A and B), and *T. equiperdum*) but not from HEK cells, synthetic controls for *T. simiae* and *T. suis* or co-endemic parasites such as *T. congolense*, *T. vivax*, *T. theileri*, and *P. falciparum* (*Figure 1B*). The species-specific *T. congolense*, *T. vivax*, *T. theileri*, *T. simiae*, and *T. suis* assays only detected their respective targeted parasite RNA or synthetic controls with no cross-reactivity with any other RNAs (*Figure 1C*). Additionally, no unspecific cross-reaction was observed with the host-derived genetic material, as assessed by testing RNA from pig blood (SHERLOCK single pan-trypanosomatid and pan-*Trypanozoon* assays in *Figure 2—figure supplement 1*).

The analytical sensitivity of each assay was quantified using serial dilutions of positive control RNAs from known numbers of parasites. For the single pan-trypanosomatid assay, the LoD was between 10 and $10^3$ parasites per ml. The pan-*Trypanozoon* and *T. theileri* assays tested 10 parasites per ml. For the *T. congolense* assay, the LoD was found between 10 and $10^3$ parasites per mL, and between $10^4$ and $10^5$ parasites per ml for the *T. vivax* test (*Figure 2A*). The sensitivity of *T. simiae* and *T. suis* tests was evaluated on synthetic positive controls, as no genetic materials were available, and both showed an LoD between 10 and 100 fM (*Figure 2B*).

Then, to improve the sensitivity of the simple pan-trypanosomatid assay, crRNA1 was tested in combination with crRNA3 for target recognition, as both crRNAs fall within the same RPA amplicon A (*Figure 3A*). Comparative analyses between the two versions were performed and multiplexing was shown to increase the performance of the assay, especially for the lower dilutions evaluated (100, 500, and $10^3$ p/ml) by increasing the fold-change readouts of these positive controls by 1-, 2.5-, and 3-fold-change units, respectively (*Figure 3—figure supplement 1*). Using results obtained on the blood samples from pigs in Côte d'Ivoire (see next section), we compared the two versions of the pan-trypanosomatid SHERLOCK assays to determine the posterior distribution of sensitivity estimates. We applied a Monte Carlo method (Gibbs sampling) proceeding in an iterative chain using the infection frequency, specificity, and sensitivity of the two tests to update the infection status. We determined the posterior sensitivity distribution of the simple versus multiplex pan-trypanosomatid SHERLOCK assays. Multiplexing crRNAs increased the median sensitivity from 66.1% to 74.7%, while keeping specificity above 80% (*Figure 3B*). Furthermore, a higher concordance was observed between the results obtained on the same samples with the pan-*Trypanozoon* SHERLOCK assay and the multiplexed pan-trypanosomatid SHERLOCK assay (25.7% of double positives), as compared to the simple version (7.8% of double positives) (*Figure 3C*). Given its improved detection efficiency, the multiplex version of the 18 S rRNA pan-trypanosomatid assay was subsequently used exclusively and referred to as the multiplex pan-trypanosomatid assay.

To determine a positivity threshold value (fold-change cut-off) for each SHERLOCK assay, receiver operating characteristic (ROC) curve analyses were performed and used to evaluate each

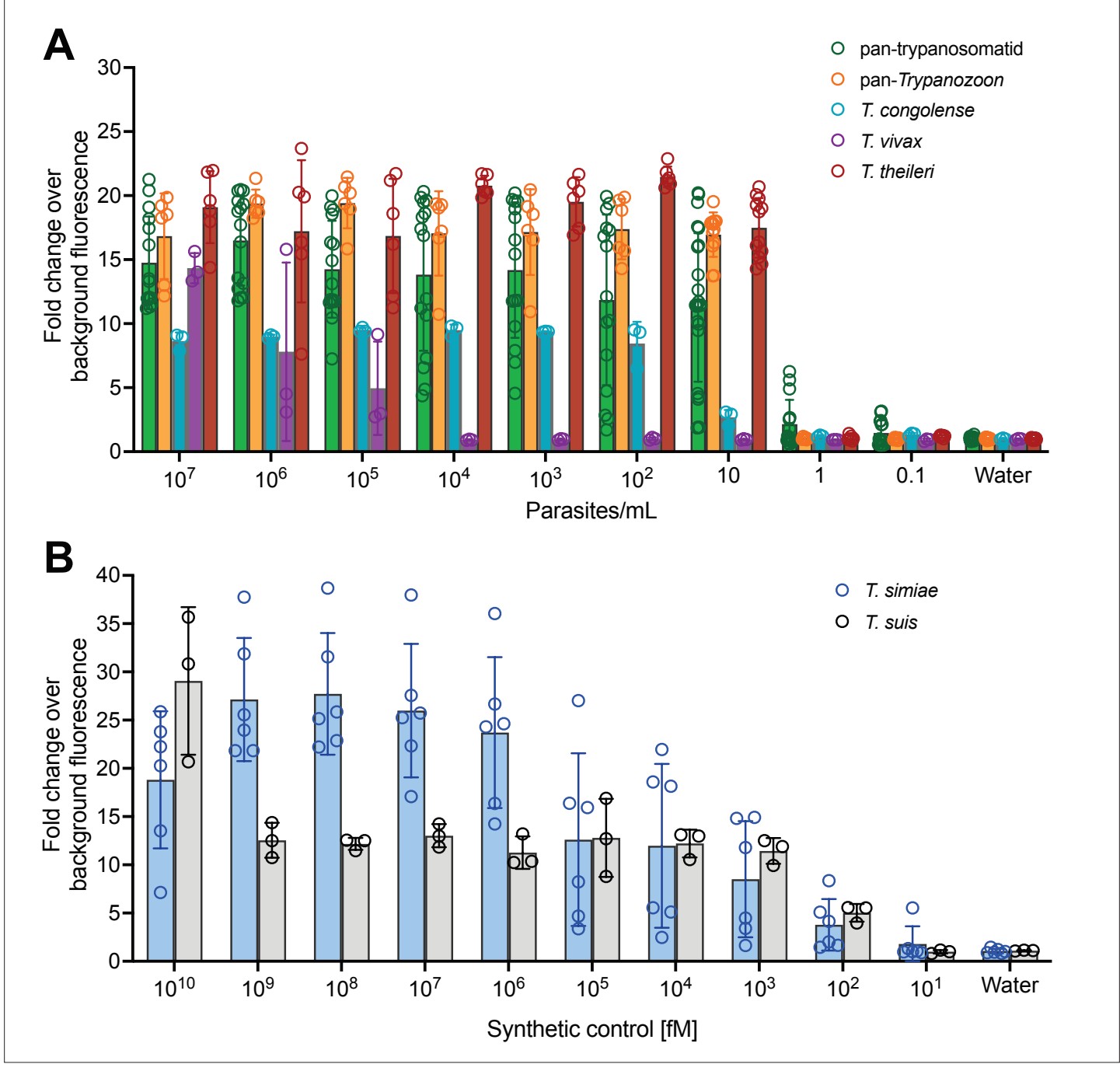

**Figure 2.** Evaluation of the sensitivity of SHERLOCK assays targeting trypanosome species causing animal African trypanosomosis (AAT). (**A**) Limit of detection (LoD) assays for the single pan-trypanosomatid, pan-*Trypanozoon*, *T. vivax*, *T. congolense*, and *T. theileri* assays were performed using RNAs from bloodstream parasites diluted to obtain equivalent concentrations of parasites per ml, ranging from $10^7$ to 0.1 parasites/ml. (**B**) LoD of *T. simiae* and *T. suis* assays were performed on consecutive dilutions of the synthetic controls, ranging from $10^{10}$ to $10^1$ fM. $N = 24$ for pan-trypanosomatid assay, $N = 12$ for pan-*Trypanozoon* and *T. theileri* assays, $N = 6$ for *T. simiae* assay, and $N = 3$ for *T. congolense*, *T. vivax*, and *T. suis* assays. Error bars represent standard deviations from the mean of the replicates; open circle represents each replicate.

The online version of this article includes the following source data and figure supplement(s) for figure 2:

**Source data 1.** LoD assays for the single pan-trypanosomatid, pan-Trypanozoon, *T. vivax*, *T. congolense*, and *T. theileri* SHERLOCK.

**Source data 2.** LoD assays for the *T. simiae* and *T. suis* SHERLOCK.

**Figure supplement 1.** Background cross-reactivity of the 18S pan-trypanosomatid (**A**) and 18S pan-*Trypanozoon* (**B**) SHERLOCK assays using genetic material extracted from pig blood as input.

*Figure 2 continued on next page*

*Figure 2 continued*

**Figure supplement 1—source data 1.** Background cross-reactivity assays for the 18S pan-trypanosomatid and 18S pan-Trypanozoon SHERLOCK.

**Figure supplement 2.** Threshold determination for all new SHERLOCK4AAT assays.

**Figure supplement 2—source data 1.** Data for threshold determination for all new SHERLOCK4AAT assays.

following analytical sensitivity and specificity. The positivity threshold values were 6.050 for the pan-trypanosomatid test, 11.70 for the pan-*Trypanozoon* test, 1.923 for the *T. congolense* test, 1.831 for the *T. vivax* test, 1.805 for the *T. theileri* test, 1.546 for the *T. simiae* test, and 2.609 for the *T. suis* test (*Figure 2—figure supplement 2*). Subsequently, for a given test, any sample with a fold-change value above the threshold was considered positive for this test. Unsurprisingly, assays designed against multiple copy genes (*18S rRNA* and *GAPDH*) showed a greater sensitivity than those targeting single copy genes (*IFX*). Taken together, all the validated tests were able to specifically detect the RNA targets for which they were designed, making it possible to distinguish between closely related trypanosomatid parasite species. We then applied this novel SHERLOCK4AAT toolbox was applied to the analysis of field samples in a molecular eco-epidemiological survey.

## Application of the SHERLOCK4AAT toolbox to eco-epidemiological surveys in pigs

In transmission foci of Western Africa, domestic pigs can host multiple trypanosome species. To investigate this, the SHERLOCK4AAT toolbox was used to survey trypanosomes circulating in domestic pigs from hypo-endemic and historical foci in Côte d'Ivoire and Guinea. In Côte d'Ivoire, blood samples were collected from 224 free-ranging pigs across the rural districts of Bonon, Bouaflé, Sinfra, Vavoua, and Brobo (*Figure 4A*). In the N'zérékoré prefecture of Forested Guinea, 200 farm pigs were sampled, with a focus on organized farming communities (*Figure 5A*). In addition to the SHERLOCK4AAT toolbox assays, all samples were also screened for *T. b. gambiense* using the previously described TgsGP SHERLOCK assay (*Sima et al., 2022*).

In total, 74.6% [68.9, 80.3] (167/224) of the samples tested positive in at least one SHERLOCK assay in Côte d'Ivoire (*Figure 4B*), and 49.5% [42.6, 56.9] (99/200) in Guinea (*Figure 5B*) representing a high apparent prevalence of trypanosome infections in these two populations of pigs. In parallel, 47.3% [40.8, 53.8] (106/224) of the pigs from Côte d'Ivoire tested positive using at least one parasitological method (buffy coat examination, mAECT and/or isolation in mice; results detailed in a separate study, manuscript in preparation). No systematic parasitological data was available for Guinean samples. The difference in positivity rates between SHERLOCK detection and parasitological approaches on Ivoirian samples was likely due to the higher sensitivity of SHERLOCK as compared to classical methods. Indeed, we compared three SHERLOCK assays (multiplex pan-trypanosomatid, pan-*Trypanozoon*, and *T. congolense*-specific SHERLOCK assays) to the three parasitological tests (buffy coat examination, mAECT and/or isolation in mice) to determine the posterior distribution of sensitivity estimates. A Monte Carlo method (Gibbs sampling) was used by proceeding in an iterative chain using the infection frequency, specificity, and sensitivity of the tests to update the infection status. This analysis suggested that, in terms of sensitivity, the combination of SHERLOCK4AAT assays may compete with combined parasitological methods (i.e., median sensitivity 91.4% vs 84.3%) while controlling median specificity above 75% (*Figure 4—figure supplement 1*). Importantly, in contrast to parasitological tests, SHERLOCK4AAT allowed for the identification of the trypanosome species involved in more than half of these infections.

Using the multiplex pan-trypanosomatid assay, 41.1% [34.6, 47.6] of the samples from Côte d'Ivoire (90/219) and 29.5% [23.2, 35.8] from Guinea (59/200) were positive (*Figures 4C and 5C*). The pan-*Trypanozoon* assay identified 38.0% [31.6, 44.4] (85/224) and 39.0% [32.2, 45.8] (78/200) of samples in each country, respectively (*Figures 4C and 5C*). As there is no specific test for *T. b. brucei*, samples that were positive in the pan-*Trypanozoon* assay but negative in the TgsGP assay were presumed to be *T. b. brucei* infections. Only one sample tested positive in the *T. b. gambiense*-specific assay in each country (*Figures 4B and 5B*). The TgsGP-positive sample from the district of Bonon, Côte d'Ivoire was negative for all other SHERLOCK assays, as well as in all parasitological tests. In contrast, the TgsGP-positive sample from Guinea was positive in the multiplex pan-trypanosomatid SHERLOCK assay, as well as in the CATT and mAECT tests, but it was negative in the pan-*Trypanozoon* SHERLOCK assay.

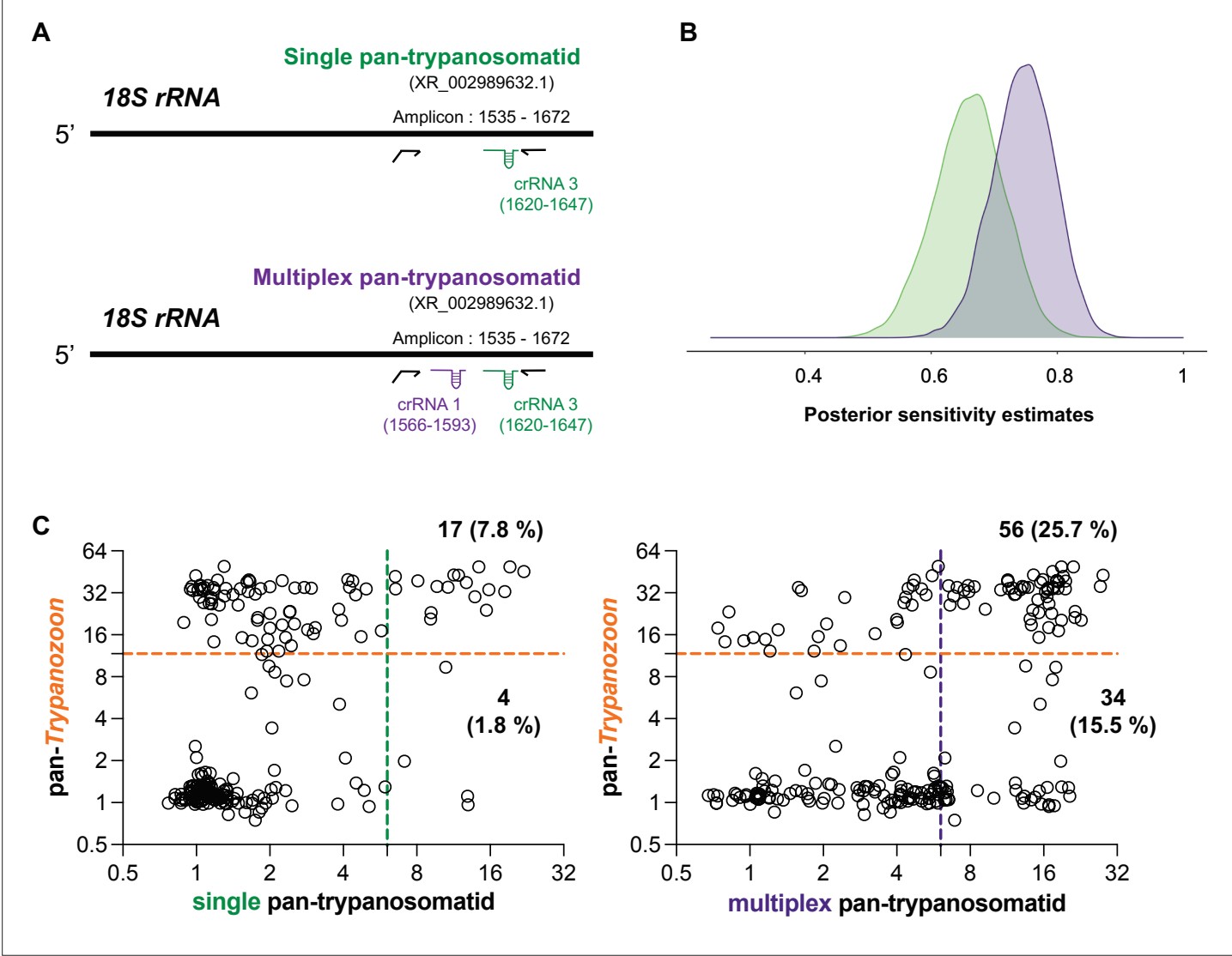

**Figure 3.** Comparison of the single and multiplex 18S pan-trypanosomatid SHERLOCK assays efficiency. (**A**) Schematic of the single (one crRNA) and multiplex (two crRNAs) pan-trypanosomatid SHERLOCK assays designed using the *18S rRNA* gene as a target. Gene-specific accession number in brackets below the assay name. Details of the amplified gene regions are provided (amplicon). RPA primers are represented by black arrows; the forward primer extension represents the T7 polymerase promoter sequence. The regions targeted by the crRNAs are indicated in the diagram. (**B**) Posterior sensitivity estimates of the single (green) and multiplex (purple) pan-trypanosomatid SHERLOCK assays on the 224 pig blood samples from Côte d'Ivoire presented in *Figure 4*. Estimates were obtained with the Monte Carlo method (Gibbs sampling) proceeding in an iterative chain. (**C**) Results of pan-*Trypanozoon* SHERLOCK assay plotted against results of the single (left panel) and multiplex (right panel) 18S pan-trypanosomatid SHERLOCK assays on pig samples from Côte d'Ivoire. The coloured dashed lines represent the detection thresholds set for each of the tests: single pan-trypanosomatid (green), multiplex pan-trypanosomatid (purple), and pan-*Trypanozoon* (orange). Numbers and proportions (%) of double positive samples are indicated at the top right of each graph and only positive for pan-trypanosomatid in the bottom right part of each graph.

The online version of this article includes the following source data and figure supplement(s) for figure 3:

**Source data 1.** Results of the pan-Trypanozoon, single and multiplex 18S pan-trypanosomatid SHERLOCK assays on pig samples from Côte d'Ivoire.

**Figure supplement 1.** Comparison of the limit of detection of the single and multiplex 18S pan-trypanosomatid SHERLOCK assays.

**Figure supplement 1—source data 1.** LoD assays for the single and multiplex 18S pan-trypanosomatid SHERLOCK.

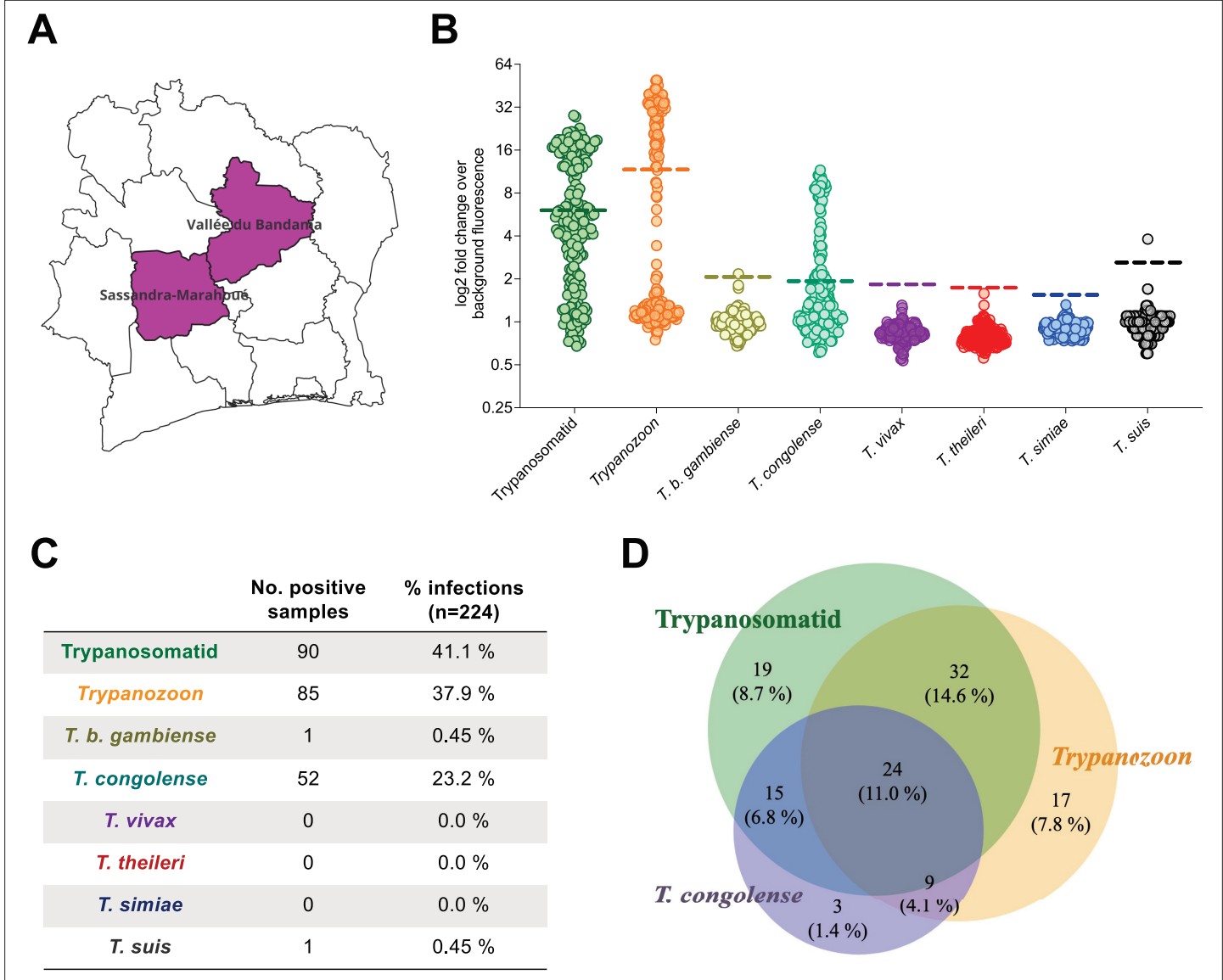

**Figure 4.** Diversity of blood trypanosomatids among free-range pigs from Côte d'Ivoire assessed by the SHERLOCK4AAT toolbox. (**A**) Map of the Côte d'Ivoire, the region where the samples were collected is highlighted in purple. (**B**) Results of the SHERLOCK4AAT screening of 224 pig samples collected in Côte d'Ivoire. The coloured dashed lines represent the detection thresholds set for each assay. Each colour represents a specific test: multiplex pan-trypanosomatid (green), pan-*Trypanozoon* (orange), *T. b. gambiense* (yellow), *T. congolense* (light green), *T. vivax* (purple), *T. theileri* (red), *T. simiae* (blue), and *T. suis* (black). (**C**) Table summarizing the number of positive samples as well as the percentage (%) positive infections for each test, calculated on the total number of samples analysed. The total number of samples analysed was 219 for the multiplex pan-trypanosomatid assay, 173 for the *T. vivax*-specific assay, and 224 for all other SHERLOCK assays. (**D**) Venn diagram representing the crossover between the positive samples analysed with the three tests (multiplex pan-trypanosomatid (green), pan-*Trypanozoon* (orange) and *T. congolense* (purple)) giving the highest prevalence. The percentage of the total is shown in brackets below the number of samples constituting each fraction.

The online version of this article includes the following source data and figure supplement(s) for figure 4:

**Source data 1.** Results of the SHERLOCK4AAT screening of 224 pig samples collected in Côte d'Ivoire.

**Figure supplement 1.** Posterior sensitivity estimates of three SHERLOCK assays (multiplex pan-trypanosomatid, pan-*Trypanozoon*, and *T. congolense*-specific SHERLOCK assays, red curve) to the three parasitological tests (buffy coat examination, mAECT and /or isolation in mice, orange curve) performed in parallel on the 224 pig blood samples from Côte d'Ivoire.

**Figure supplement 1—source data 1.** Posterior sensitivity estimates of three SHERLOCK assays to the three parasitological tests.

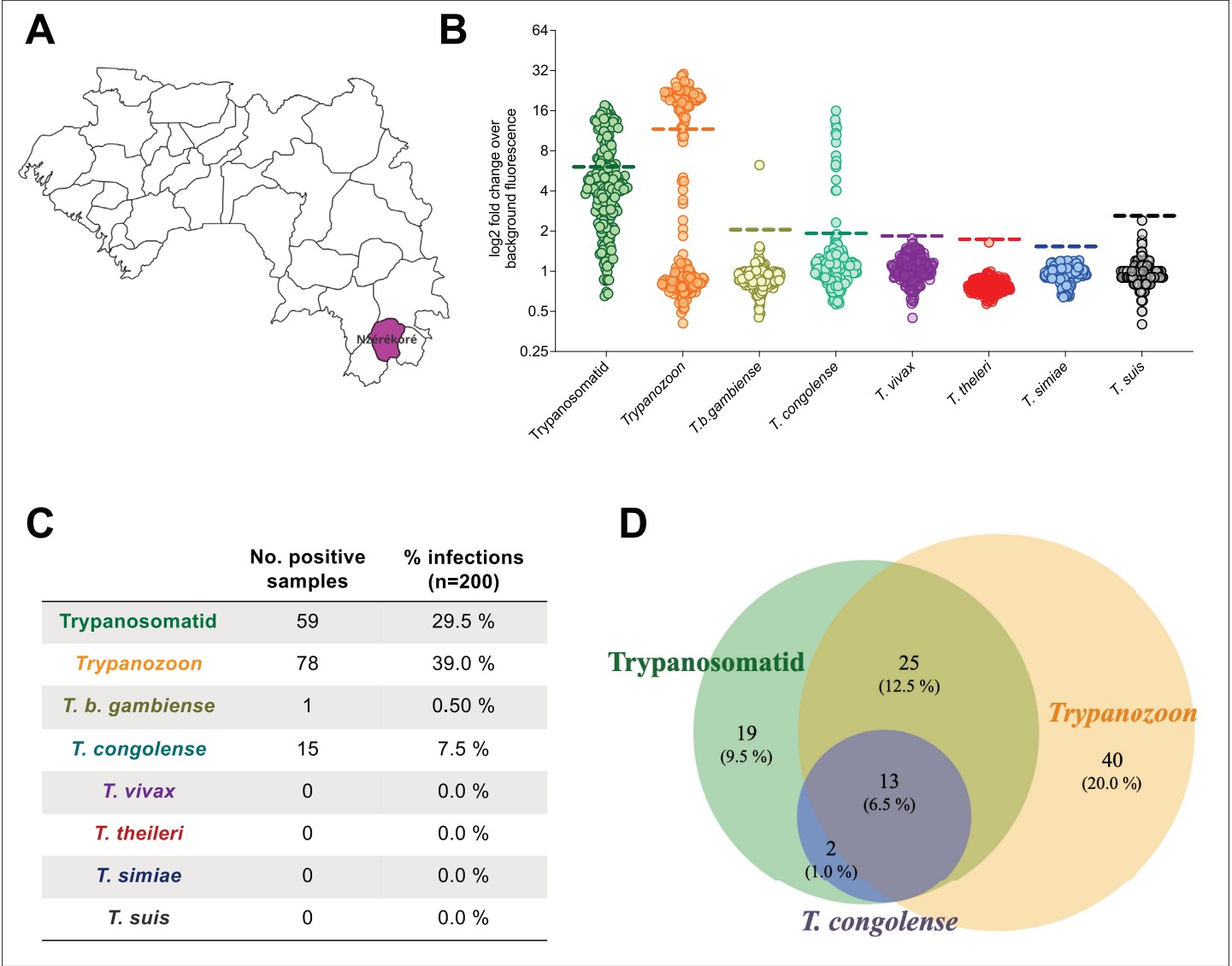

**Figure 5.** Diversity of blood trypanosomatids among farm pigs from Guinea assessed by the SHERLOCK4AAT toolbox. (**A**) Map of Guinea showing the N'zérékoré prefecture where samples were collected in pink. (**B**) Results of the SHERLOCK4AAT screening of 200 pig samples collected in Guinea. The coloured dashed lines represent the detection thresholds set for each of the tests. Each colour represents a specific test: multiplex pan-trypanosomatid (green), pan-*Trypanozoon* (orange), *T. b. gambiense* (yellow), *T. congolense* (light green), *T. vivax* (purple), *T. theileri* (red), *T. simiae* (blue), and *T. suis* (black). (**C**) Table summarizing the number of positive samples as well as the percentage (%) positive infections for each test, calculated on the total number of samples analysed (*n* = 200). (**D**) Venn diagram representing the crossover between the positive samples analysed with the three tests (multiplex pan-trypanosomatid (green), pan-*Trypanozoon* (orange), and *T. congolense* (purple)) giving the highest prevalence. The percentage of the total is shown in brackets below the number of samples constituting each fraction.

To further explore trypanosome diversity, species-specific SHERLOCK assays for AAT were applied to the samples. In total, 23.2% [17.7, 28.7] (52/224) of the samples in Côte d'Ivoire and 7.5% [3.8, 11.2] (15/200) in Guinea were positive for *T. congolense*. Additionally, a single sample from Côte d'Ivoire tested positive in the *T. suis*-specific SHERLOCK assay. Furthermore, no RNA from *T. vivax* or *T. theileri* was detected with the IFX SHERLOCK assay.

Interestingly, a substantial number of co-infections were observed. In total, 24/219 individuals (11.0 ± 4.1%) in Côte d'Ivoire and 13/200 in Guinea (6.5 ± 3.4%) tested positive in the multiplex pan-trypanosomatid, the pan-*Trypanozoon*, and the *T. congolense*-specific assays. These triple positive results indicate co-infections involving both a *Trypanozoon* species (likely *T. b. brucei*) and *T. congolense* (**Figures 4D and 5D**). In addition, 9/219 samples (4.1 ± 2.6%) from Côte d'Ivoire tested positive in the pan-*Trypanozoon* and the *T. congolense* assays, which also represented co-infections

with *T. brucei s.l.* and *T. congolense*. Overall, 15.2% [10.5, 19.9] of the samples from Côte d'Ivoire showed co-infections with a *Trypanozoon* species and *T. congolense*. Additionally, 15 samples from Côte d'Ivoire tested positive in both the multiplex pan-trypanosomatid assay and the *T. congolense*-specific assay or were positive only for *T. congolense*. In Guinea, two samples tested positive in both the *T. congolense*-specific and multiplex pan-trypanosomatid SHERLOCK assays, indicating *T. congolense* mono-infections (*Figures 4D and 5D*).

Furthermore, 56 samples (25.6 ± 5.8%) in Côte d'Ivoire and 38 samples (19.0 ± 5.4%) in Guinea tested positive in both the multiplex pan-trypanosomatid and the pan-*Trypanozoon* SHERLOCK assays. In contrast, 17 samples from Côte d'Ivoire and 40 from Guinea tested positive in the pan-*Trypanozoon* assay but negative in the multiplex pan-trypanosomatid assay, likely due to the lower sensitivity of the latter. Lastly, 19 samples (8.7 ± 3.7%) in Côte d'Ivoire and 19 samples (9.5 ± 4.1%) in Guinea tested positive in the multiplex pan-trypanosomatid assay only (*Figures 4D and 5D*). The absence of positivity in the *T. congolense*, *T. simiae*, and *T. suis*-specific assays may be due to their lower sensitivity, or to infections with other non-targeted trypanosome species.

## Discussion

### The SHERLOCK4AAT toolbox, a new panel of highly sensitive and specific high-throughput assays for detecting animal trypanosomes

Species identification is essential for assessing both the distribution of prominent parasites impacting veterinary health and spotting reservoirs for human-infective trypanosomes. In a context where symptoms are often mild and unspecific, especially in hosts like pigs, accurate species-specific detection methods are essential. Ideally, diagnostic assays should be able to analyse both fresh and stored (ambient temperature) samples to determine parasites circulating within a specific area.

In this study, conserved and species-specific targets have been used to adapt SHERLOCK for the detection of most members of the Trypanosomatidae family (Kinetoplastea class) (*Supplementary file 4*) of veterinary importance (pan-trypanosomatid), and species-specific assays to discriminate between the six most common animal trypanosome species in the study regions: *Trypanozoon* (*T. b. brucei*, *T. evansi*, and *T. equiperdum*), *T. congolense* (Savannah, Forest Kilifi, and Dzanga-sangha), *T. vivax*, *T. theileri* (TthI, TthII, and TthIII), *T. simiae*, and *T. suis*. The design of both broad and species-specific assays was based primarily on sequences of the *18S rRNA* and *GAPDH* genes which, as well as being conserved across species (*Figure 1—figure supplement 1*), are highly expressed and contain regions that allowed for species discrimination. However, for *T. vivax*, the single-copy *IFX* gene was selected for its high specificity (*Romero-Ramirez et al., 2022*). The first simple version of our pan-trypanosomatid assay employed a single crRNA guide and was previously used to evaluate blood samples from gHAT seropositive individuals with a specificity of 93% (95.4–98.6) (*N'Djetchi et al., 2024*). To increase the performances of this assay, we added a second crRNA guide into the reaction mix to target the same RPA amplicon during the SHERLOCK reaction, resulting in an increased test sensitivity (*Figure 3*, *Figure 3—figure supplement 1*). For *Trypanozoon* species, no sequences were identified to distinguish *T. b. brucei*, *T. evansi* (A and B), and *T. equiperdum* as these parasites cannot be easily differentiated using the *18S rRNA* gene. In the future, other genes such as *RoTat1.2 VSG* or *ESAG6/7* could be explored to discriminate *T. evansi* from other *Trypanozoon* by SHERLOCK (*Claes et al., 2004*; *Holland et al., 2001*). Overall, in silico and in vitro analyses enabled us to assess the analytical specificities of these new SHERLOCK assays, confirming that the targets selected were sufficiently specific to be used as diagnostics for the species and groups of species concerned.

Using parasite RNAs, the LoD of the SHERLOCK assays was estimated to range between 10 and $10^3$ parasites/ml for the pan-trypanosomatid assay, 10 parasites/ml for the pan-*Trypanozoon* and *T. theileri* assays, from 10 to $10^3$ parasites/ml for *T. congolense* and $10^4$–$10^5$ parasites/ml for *T. vivax*. Most PCR-derived molecular techniques currently used for detecting animal trypanosomes have a detection limit of 1–10 parasites/ml using genus- or species-specific primers, 50 parasites/ml using pan-species ITS1 primers (*Desquesnes et al., 2022a*) or 5 parasites/ml using 18S rRNA-targeting primers (*Deborg-graeve et al., 2006*). The ROC analyses of the *T. brucei*, *T. congolense*, and *T. vivax* SHERLOCK assays (*Figure 2—figure supplement 2C*) also showed sensitivity and specificity comparable to those of the recently developed 7SL sRNA diagnostic test for Animal trypanosomosis (*Contreras Garcia et al., 2022*). Therefore, in total, the analytical sensitivities obtained by the pan-trypanosomatid,

pan-*Trypanozoon*, *T. theileri*, and *T. congolense* SHERLOCK assays were comparable with those of other existing molecular tools. Yet, a significant difference in favour of using SHERLOCK4AAT is that it detects RNAs, probing for active infections. The *T. vivax* SHERLOCK assay, targeting the single-copy *IFX* gene (*Autheman et al., 2021*), did not meet the sensitivity requirements for the molecular diagnosis of animal trypanosomes and would require the identification of another target. As no genetic material from parasites was available for developing *T. simiae* and *T. suis* SHERLOCK assays, LoDs were determined using synthetic RNA controls. Both assays were able to detect 10–100 fM of the synthetic controls, far from the detection capacity of SHERLOCK, which can detect synthetic targets at a concentration of 2 aM in optimal conditions (*Gootenberg et al., 2017*).

In the current version of the SHERLOCK4AAT toolbox, each assay can be run in 2 hr 30 min, in a two-step reaction, at 2.5 € (exclusive of initial sample processing) (*Sima et al., 2022*). All tests are optimized for high throughput screening with a fluorescence readout; hence, they can be performed in standard lab conditions by a technician trained for basic molecular biology. SHERLOCK4AAT was not developed for direct case diagnosis in the field, but rather for eco-epidemiological surveillance at large scale. Considering the absence of trypanosome species-specific treatment for domestic animals, the future development of a RDT for AAT would only include the pan-trypanosomatid target or equivalent. This would, however, require further technical optimizations (TNA extraction, sensitivity, LFA, etc.).

## SHERLOCK4AAT profiling in Guinea and Côte d'Ivoire

Trypanosome species identification can be used to finely map the risk of AAT parasite transmission in these areas. After cattle, pigs are the main domestic hosts infected by animal trypanosomes, but they can also be reservoirs of *T. b. gambiense*. *T. b. gambiense* infections in pigs are frequently reported in sub-Saharan Africa (*Ebhodaghe et al., 2018*; *Okello et al., 2022a*) and these domestic animals likely represent an underestimated reservoir of human-infective parasites (*Büscher et al., 2018*; *Mehlitz and Molyneux, 2019*). This is all the more important than pigs are often raised in close proximity to humans, increasing the risk of animal to human trypanosome transmission, reinforcing our efforts to integrate the detection of both animal and human parasites in domestic pigs in a One Health perspective (*Rotureau et al., 2022*).

In blood samples from Ivorian pigs, analyses with the SHERLOCK4AAT toolbox revealed 37.9% positive samples for *Trypanozoon* RNAs and 23.2% positive samples for *T. congolense* RNAs, with 15.2% samples positive for both tests. In addition, no sample was positive for *T. vivax*, *T. theileri* or *T. simiae*, but one sample was positive in the *T. suis* assay and another positive for *T. b. gambiense*. These findings are consistent with the most recent known distribution of animal trypanosomes in free-ranging pigs raised in Côte d'Ivoire's historical gHAT foci. The use of conventional molecular diagnostics on 97 free-range pigs resulted in an overall positivity rate of 57% for *Trypanozoon* DNA (PCR-TBR) and 24% for *T. congolense forest* DNA (PCR-TCF), with 14% of pigs positive in both PCRs and none positive for *T. vivax* (PCR-TVW) (*N'Djetchi et al., 2017*; *Traoré et al., 2021*). The epidemiological status of AAT in Guinea is unclear, as no large-scale molecular studies have been published so far. The blood samples from farmed pigs analysed here were collected in the prefecture of N'zérékoré, a former endemic focus of gHAT. Results showed a positivity rate of 39.0% for the detection of *Trypanozoon* RNAs, like that observed in Côte d'Ivoire, but a lower rate of positivity for *T. congolense* RNAs (7.5%), with 6.5% of samples positive for both tests. In addition, one sample tested positive for *T. b. gambiense,* but no *T. vivax*, *T. simiae*, *T. theileri* nor *T. suis* was detected. The lack of any samples positive for *T. vivax or T. theileri* in any study areas confirms that pigs are not frequent carriers of these species (*Fetene et al., 2021*). However, the blood samples positive for *T. b. gambiense* in each study site provide further evidence of the ability of pigs to carry human trypanosomes and to act as potential reservoirs. Similarly, the presence of *T. b. gambiense* genetic material in pig samples has already been suspected in Côte d'Ivoire (*Traoré et al., 2021*).

In both sites, a high prevalence of trypanosomatid parasites was observed using the multiplex pan-trypanosomatid and/or pan-*Trypanozoon* assays (42.0% in Côte d'Ivoire and 48.5% in Guinea). Having partially deconvolved the species present in these animals, there is still a small proportion (8.7% in Côte d'Ivoire and 9.5% in Guinea) of trypanosomatid parasites that remained uncharacterized by our tests on samples found only positive in the multiplex pan-trypanosomatid SHERLOCK assay. Firstly, these assays were designed by in silico determination of the most specific and conserved

region to target the entire Trypanosomatidae family, and even Kinetoplastea class, or to discriminate between species, yet the limited number of sequences available for some species in the literature and in the online databases may under-represent the real genetic diversity of these parasites in natura. In line with this hypothesis, there may also exist uncharacterized trypanosome species circulating among these reservoirs, as described in small mammals in central and East Africa (*Votýpka et al., 2022*), and that were only detected by the broad multiplex pan-trypanosomatid assay. Altogether, these results are consistent with those reported in the few available previous studies (*N'Djetchi et al., 2017*; *Traoré et al., 2021*) and confirm the effectiveness of the SHERLOCK4AAT toolbox in analysing the distribution of trypanosomes in hypo-endemic gHAT foci. During natural infections, trypanosome nucleic acids can be detected in the skin, adipose tissue and various organs even if the blood remains negative (*Amisigo et al., 2024*). Trypanosome DNA also persists in blood and skin, even after drug treatment (*Ekloh et al., 2023*). Hence, applying SHERLOCK detection on skin samples in the future would be useful for monitoring both blood and skin-dwelling parasites. The future development of multiplexed assays combined with adapted lateral flow tests would also be highly desirable for reaching field-applicable point-of-care molecular diagnostic versions of the tests useable in the field. The absence of point of care diagnostic tests for AAT is currently a major hurdle for its surveillance and control (*Desquesnes et al., 2022b*).

*T. b. gambiense* is suspected to circulate among domestic animals raised in close contact to humans, such as pigs. Here, *T. b. gambiense* has been detected in one animal in each study site. Considering the resulting low prevalence of *T. b. gambiense* infections in these two populations of domestic pigs (0.45 ± 0.9% in Côte d'Ivoire to 0.50 ± 1.0% in Guinea) and the short life expectancy of domestic pigs reared for commercial purposes in these two rural areas (6–8 months), these animals do not likely represent a significant risk to the elimination of gHAT. This is apparently confirmed by the absence of new human cases as well as by the low abundance of tsetse flies in the two sites (article in preparation). However, because of their proximity to humans and their easy access for frequent sampling, pigs could be used as sentinels to monitor *T. b. gambiense* circulation, possibly with the SHERLOCK4AAT toolbox.

## Materials and methods
### Mice infection and parasite purification
Twenty-four hours prior to infection with trypanosome parasites, 8-week-old female BALB/c mice (Charles River Laboratory) were immunosuppressed by intraperitoneal injection of 0.1 ml Endoxan 500 mg (Baxter). *T. congolense* Savannah subgroup IL1180 strain (Tanzania, Serengeti 1971, STIB212), *T. vivax* Magnan strain (Burkina Faso, Magnan 2011), and *T. theileri* (Nederland, Utrecht 2003, UNUT) parasites (INTERTRYP unit's cryobank) were inoculated by intraperitoneal injection of 0.15 ml of the cryopreserved blood containing parasites. Two mice were infected per strain and parasitaemia was monitored daily by tail blood sampling. Parasite yields were estimated by counting in a fast read 102 counting cells (Kova). Whole blood was collected by cardiac puncture under deep isoflurane anaesthesia when parasitaemia reached $1 \times 10^8$ cells per ml. Infected blood was passed over a diethylaminoethyl (DEAE)-cellulose column and the column washed using phosphate buffer saline (PBS) supplemented with 1% glucose (PBSG). Parasites were collected in PBSG and concentrated by centrifugation (2000 rpm, 10 min, RT).

### Cell culture
*T. b. brucei* Lister 427 (MiTaT 1.2, clone 221) bloodstream form cells from Institut Pasteur (Paris, France), and *T. b. gambiense* ELIANE (LiTAR1, LiTat 1.3, Paris/52/-/-/(ELIANE) ITMAS100500) bloodstream form cells from College of Medical, Veterinary and Biological Sciences (Wellcome Trust Centre for Molecular Parasitology, Glasgow, United Kingdom) were cultured in HMI-11 medium (Thermo Fisher Scientific) supplemented with 10% fetal bovine serum (Sigma) at 37°C with 5% $CO_2$. The identity of these two cell lines was confirmed in SHERLOCK tests (*Figure 1*). Commercially available HEK 293T cells (Lonza), quality-checked negative for *Mycoplasma* contamination, were cultured in complemented dMEM (Merck) at 37°C with 5% $CO_2$.

### Epidemiological surveys
Pig blood samples were collected from the external jugular vein in heparinized tubes during two independent surveys in Côte d'Ivoire and Guinea. In Côte d'Ivoire, 224 free-ranging pigs were sampled

in May, June, and July 2021 in the Bonon and Sinfra hypo-endemic HAT foci, the Bouaflé and Vavoua historical HAT foci and the Brobo HAT-free area. In Guinea, 200 farm pigs were sampled in March 2023 in the N'zérékoré prefecture, an ancient gHAT focus. Four drops of blood were dried on 4th grade Whatman filter paper (Cytivia Life Sciences) and conserved in sealed plastic bags with silica gel at ambient temperature for several months until TNA extraction.

## Nucleic acid extraction

RNA was extracted from bloodstream form parasites (*T. b. brucei* Lister 427, *T. b. gambiense* ELIANE, *T. congolense* Savannah subgroup IL1180, *T. vivax* Magnan, and *T. theileri*), HEK 293T cells and whole blood from French cattle, pig, sheep, and goat (Bergerie de la Combe aux Loups). Bloodstream form parasites from infected mice, as well as cultured HEK 293T cells, were harvested in TRIzol (Life Technologies) and Chloroform:Isoamyl alcohol (24:1, Sigma-Aldrich) was added, followed by centrifugation at 13,000 rpm at 4°C for 15 min. The upper aqueous phase was transferred to a fresh tube containing 0.5 ml of 75% ethanol. After incubation for 10 min at room temperature, RNA was extracted using a NucleoSpin RNA column (Macherey Nagel) or a RNAeasy kit (QIAGEN) according to the manufacturer's instructions. 500 µl of blood sampled from French cattle, pig, sheep and goats was treated with 500 µl lysis buffer supplemented by 50 µl Proteinase K and incubated for 30 min at 56°C. The resulting mixture was transferred into a QIAshredder column (QIAGEN) and centrifuged at 13,000 rpm for 2 min. The flow-through was then used for RNA extraction using an RNeasy Mini kit (QIAGEN) according to the manufacturer recommendations. After extraction, qualities and quantities of RNA were evaluated using the NanoDrop One (Ozyme).

For extraction of TNA from dry blood spots (DBS), three 6 mm punches from one spot of each animal DBS were resuspended in 300 µl of lysis buffer supplemented by 30 µl of Proteinase K and incubated for 30 min at 56°C. The punches and liquid were transferred into a QIAshredder column (QIAGEN) and centrifuged at 13,000 rpm for 2 min. TNA was extracted from the flow-through using the Maxwell RSC Buccal Swab Kit (AS1640) with the Maxwell RSC Instrument (AS4500) according to the manufacturer's instructions. For each sample, all SHERLOCK assays were performed from a single TNA extraction.

## Target selection, RPA primers, and crRNA design

Alignments of the *18S rRNA*, *IFX*, and *GAPDH* genes were performed on sequences from different species (*Supplementary file 1*) using Clustal Omega (*Sievers and Higgins, 2018*). The distance matrix representing the percentage identity between each sequence was represented using GraphPad Prism software (V.[10]).

BLAST analysis (*McGinnis and Madden, 2004*) of the *18S rRNA* and *GAPDH* genes selected target sequences was performed in the NCBI standard database of using the following selection of organisms: Microsporidia (taxid:6029), Ciliophora (taxid:5878), Haemosporida (taxid:5819), Diplomonadida (taxid:5738), Symbiontida (taxid:797087), Diplonemea (taxid:191814), Euglenida (taxid:3035), Kinetoplastea (taxid:5653), Metakinetoplastina (taxid:2704647), Trypanosomatida (taxid:2704949), Trypanosomatidae (taxid:5654), and Euglenozoa (taxid:33682). The program selection was optimized for somewhat similar sequences (blastn) and in algorithm parameters, the maximum number of aligned sequences to display was set as 5000.

The presence of SNPs in the target regions was assessed by aligning sequence variants of the parasite species using Clustal Omega (*Sievers and Higgins, 2018*). Sequence variants used are listed in *Supplementary file 1*. SNPs identified in the TriTrypDB (https://tritrypdb.org) database were also taken into account in the design: NGS_SNP.Tb927_03: v5.1.921431, v5.1.921465, v5.1.921571, v5.1.921612, v5.1.921937, v5.1.922234, v5.1.922298, v5.1.922511, v5.1.922637, v5.1.922803, v5.1.923180, v5.1.959274; NGS_SNP.Tb927_07: v5.1.1951525, v5.1.1952283, v5.1.1951291; NGS_SNP.Tb927_02: v5.1.371291, v5.1.371584, and v5.1.370058.

Primers for Recombinase Polymerase Amplification (RPA) of targeted regions were designed in Primer-BLAST (*Ye et al., 2012*) using the customized parameters described by *Kellner et al., 2019*. Oligonucleotides prone to generate self-dimers or cross-primer dimers detected by Multiple Primer Analyzer (Thermo Fisher) were discarded. Primers selected for each SHERLOCK assay are listed and highlighted in *Supplementary file 2*. The T7 promoter sequence (5' GAAATTAATACGACTCACTA

TAGGG) was added at each 5′ end of the selected forward primers to enable the in vitro transcription necessary for Cas13a detection (*Kellner et al., 2019*).

## Production of crRNAs and positive controls

crRNA production was carried out as previously described (*Kellner et al., 2019*; *Sima et al., 2022*). Briefly, each crRNA DNA template, containing the spacer (specific targeted region) followed by the direct repeat (5′GATTTAGACTACCCCAAAAACGAAGGGGACTAAAAC) and the T7 promoter sequence, was in vitro transcribed using the HiScribe T7 Quick High Yield RNA Synthesis Kit (NEB) and purified using magnetic beads (Agencourt RNAClean XP). DNA template sequences used for the synthesis of each crRNA are detailed in *Supplementary file 2*.

For the production of the synthetic positive controls, in vitro transcription of the targets was performed as previously described (*Sima et al., 2022*) using RNA from the target species for each specific assay. The amount of template used to produce the positive control for each assay was as follows: 200 ng of RNA from *T. b. brucei* Lister 427 for 18S pan-trypanosomatid and 18S pan-*Trypanozoon* assays, 120 ng of RNA from *T. b. gambiense* ELIANE strain for TgSGP assay, 120 ng of RNA from *T. vivax* Magnan strain for IFX assay, 230 ng of RNA from *T. congolense* Savannah subgroup IL1180 strain for 18S *T. congolense*-specific assay and 5 ng of RNA from *T. theileri* for 18S *T. theileri*-specific assay. Due to a lack of *T. simiae* and *T. suis* genetic material, the positive controls were synthesized at Eurogentec using the in-silico predicted sequence of the target region. Primers and target regions used to produce positive controls are listed in the *Supplementary file 3*.

## SHERLOCK assays

SHERLOCK assays were performed as previously described (*Kellner et al., 2019*; *Sima et al., 2022*). RPA was performed using the TwistAmp Basic kit (TwistDx) following the manufacturer's instructions with slight modifications. To enable reverse transcription of the RPA (RT-RPA), 2.2 U of transcriptor reverse transcriptase (Roche) was added to the reaction. Specific RPA primer pairs were used for each target amplification at a concentration of 480 nM per primer, and the amplification reaction was carried out in a heating block at 42°C for 45 min (*Sima et al., 2022*). *Leptotrichia wadeii* Cas13a (*Lw*Cas13a) enzyme was produced as previously described (*Kellner et al., 2019*; *Sima et al., 2022*). Briefly, pC013-Twinstrep-SUMO-huLwCas13a plasmid (Addgene plasmid #90097) was inserted into *Escherichia coli* RosettaTM 2(DE3) pLysS competent cells for *Lw*Cas13a expression and purification (*Gootenberg et al., 2017*). Purified *Lw*Cas13a was conserved in Storage buffer (SB: Tris-HCl 1 M, NaCl 5 M, DTT 1 M and glycerol) (*Kellner et al., 2019*) in single-use aliquots at –80°C to avoid freeze–thaw cycles.

The Cas13a–crRNA-mediated target detection reaction was performed simultaneously with the in vitro transcription of the target amplified by RT-RPA. In vitro transcription of the amplified target was performed using T7 RNA polymerase (Biosearch technology). Each Cas13a detection reaction was run in triplicate on a TECAN plate reader (INFINITE F200 PRO M PLEX) and fluorescence readouts were collected every 10 min for 3 hr. All reactions were carried out with the inclusion of a negative control template (NCT) adding nuclease-free water as input, and a positive control template (PCT), here the in vitro produced positive control for each specific SHERLOCK. For screening samples from epidemiological surveys, the same protocol as described above was used, except a TECAN plate reader (INFINITE F200 PRO Option Infinite F Nano +) was used and the assay was performed without replicates. Optimizations were carried out either at Institut Pasteur in Paris (France) or at INTERTRYP unit in Montpellier (France), while analysis of samples from epidemiological surveys was carried out at Institut Pasteur of Guinea.

## Data representation and statistical analyses

Fold-change over background fluorescence was used to represent fluorescence readouts for all SHERLOCK assays. Background fluorescence is given by the NCT evaluation, where water is used as input material for the RPA reaction. For each test, sample fluorescence obtained after 3 hr of reading was divided by NCT fluorescence at the same time,

$$FC = \frac{Fsample_{t=3h}}{Fnct mean_{t=3h}},$$

where *FC* represents Fold-Change over background fluorescence, *Fsample t = 3h* represents sample fluorescence obtained after 3 hr and *Fnct mean t = 3h* represents NCT fluorescence after 3 hr of reactions.

Analytical characteristics including threshold, sensitivity, and specificity of each SHERLOCK assay were evaluated using ROC curves calculation comparing NCT versus PCT at controlled dilutions of parasites or hosts' genetic materials. Results obtained from SHERLOCK-specific assays on targeted parasites were gathered in a positive sample column, while results obtained from NCT, hosts and non-targeted parasites analysed with the assay were placed in a negative sample column for ROC curve calculation and threshold evaluation. ROC curves were calculated using the Wilson/Brown method with a 95% confidence interval and threshold values were selected based on the Youden's index (sensitivity + specificity – 1), that defines an optimal threshold as the one that maximizes the index of classification accuracy (*Brown et al., 2001*; *Schisterman et al., 2005*). The threshold of positivity for each SHERLOCK assay was selected from ROC analyses as the value corresponding to the maximum specificity (100%). The LoD of each SHERLOCK reaction was assessed using specific tests on serial dilutions of target parasite RNA. Dilutions ranged from $5 \times 10^3$ pg/µl (equivalent to $10^7$ parasites/ml) to $5 \times 10^{-5}$ pg/µl (equivalent to 0.1 parasites/ml). All plots and statistical analyses were performed with GraphPad Prism software (V.10). The Venn diagrams were made based on the Venny model by Oliveros J.C. (BioinfoGP, CNB-CSIC).

Maps were generated with the QGIS software version 3.42.0 with backgrounds retrieved from data. gouv.fr (OSM2IGEO – Ivory Coast – SHP – EPSG:4326 for Côte d'Ivoire, and OSM2IGEO – Guinea – SHP – EPSG:4326 for Guinea).

In the absence of reliable infection status in the field data from pigs, the posterior distributions of performance estimates, such as sensitivity (Se), were jointly estimated with a Bayesian latent class model (*Joseph et al., 1995*). The model takes boolean outcomes (either positive or negative) of several tests as input. This Monte Carlo method operates based on a Gibbs sampling process. The model iteratively updates both the test performance parameters (sensitivity, specificity, and frequency of positive individuals in sample collection) and the latent infection status. A flat prior was used. After visually controlling that stationarity of the chain was quickly reached, we discarded the first 150 burn-in iterations and calculated posterior densities based on >10,000 iterations. 95% confidence intervals were calculated for each proportion to quantify the uncertainty of the estimates, providing a scientific range within which the population proportions are likely to be accurate (*Brown et al., 2001*).

## Acknowledgements

The authors thank Annette MacLeod (University of Glasgow, Glasgow, UK) for providing RNA from *T. b. rhodesiense*, Nick Van Reet and Philippe Büscher (Institute of Tropical Medicine [ITM], Antwerp, Belgium) for providing RNA from *T. equiperdum*, *T. evansi* types A and B, Gerald Spaeth and Artur Scherf (Institut Pasteur, Paris, France) for providing RNA from *Leishmania major* and *Plasmodium falciparum* and John Kelly (The London School of Hygiene & Tropical Medicine, London, UK) for providing RNA from *T. cruzi*. We warmly thank all the colleagues at the University Jean Lorougnon Guédé (Daloa, Côte d'Ivoire) and at the Institut Pierre Richet (Bouaké, Côte d'Ivoire) as well as all the health workers in the districts of Sinfra and Bonon (Côte d'Ivoire) and N'zérékoré (Guinée) for their strong support in the field. This project was funded by the Agence Nationale pour la Recherche (ANR-21-CE17-0022-01 SherPa), the Programme Investissement d'Avenir of the French Government through the two Laboratoires d'Excellence, ANR-10-LABX-62-IBEID and ANR-11-LABX-0024-PARAFRAP, and the Pasteur Network (Bourse Calmette-Yersin to AC).

## Additional information

### Funding

| Funder | Grant reference number | Author |
|---|---|---|
| Agence Nationale de la Recherche | ANR-21-CE17-0022-01 SherPa | Brice Rotureau |

| Funder | Grant reference number | Author |
| --- | --- | --- |
| Agence Nationale de la Recherche | ANR-10-LABX-62-IBEID | Brice Rotureau |
| Agence Nationale de la Recherche | ANR-11-LABX-0024-PARAFRAP | Brice Rotureau |
| Institut Pasteur | Bourse Calmette-Yersin | Aïssata Camara |

The funders had no role in study design, data collection, and interpretation, or the decision to submit the work for publication.

## Author contributions

Roger Eloiflin, Elena Pérez-Antón, Aïssata Camara, Conceptualization, Data curation, Formal analysis, Investigation, Visualization, Methodology, Writing – original draft; Annick Dujeancourt-Henry, Investigation, Methodology; Salimatou Boiro, Mélika Barkissa Traoré, Yann Le Pennec, Bakary Doukouré, Abdoulaye Dansy Camara, Moïse Kagbadouno, Investigation; Martial N Djetchi, Investigation, Writing – review and editing; Mathurin Koffi, Dramane Kaba, Mamadou Camara, Resources, Supervision, Project administration; Pascal Campagne, Formal analysis, Visualization, Methodology, Writing – review and editing; Vincent Jamonneau, Resources, Supervision, Funding acquisition, Investigation, Project administration, Writing – review and editing; Sophie Thévenon, Resources, Supervision, Project administration, Writing – review and editing; Jean-Mathieu Bart, Supervision, Funding acquisition, Investigation, Methodology, Project administration, Writing – review and editing; Lucy Glover, Brice Rotureau, Conceptualization, Resources, Supervision, Funding acquisition, Validation, Investigation, Methodology, Writing – original draft, Project administration

## Author ORCIDs

Roger Eloiflin ⓘ https://orcid.org/0000-0001-5927-5592
Mathurin Koffi ⓘ https://orcid.org/0000-0001-7369-2106
Lucy Glover ⓘ https://orcid.org/0000-0001-7191-6890
Brice Rotureau ⓘ https://orcid.org/0000-0003-0671-8999

## Ethics

The only reliable method available to maintain and amplify field-isolated animal trypanosomes is to experimentally infect laboratory animals, here mice. For this study, experimental infections were carried out at UMR INTERTRYP and were approved by the regional ethics committee for animal experimentation as part of the project APAFIS #34149-2020091118136527v5 and authorized by the French Ministry for Higher Education and Research. In Guinea, ethical approval was obtained from the Comité National d'Ethique pour la Recherche en Santé (CNERS) under agreement number 102/CNERS/19. In Côte d'Ivoire, sample collection was conducted within the framework of epidemiological surveillance activities supervised by the HAT National Elimination Program (HAT-NEP). No ethical statement is required by local authorities for domestic animal sampling. Any veterinarian may carry out blood sampling on domestic animals, with the authorization of the owner, as it is performed during prophylaxis or diagnostic campaign. Breeders gave their consent for animal sampling after being informed of the objectives of the study. For pig care, venous sampling was performed by a veterinary of the Laboratoire National d'Appui au Développement Rural (Ministry of Agriculture).

Reviewer #1 (Public review): https://doi.org/10.7554/eLife.106823.3.sa1
Reviewer #2 (Public review): https://doi.org/10.7554/eLife.106823.3.sa2
Reviewer #3 (Public review): https://doi.org/10.7554/eLife.106823.3.sa3
Author response https://doi.org/10.7554/eLife.106823.3.sa4

# Additional files

## Supplementary files

Supplementary file 1. Accession numbers of organisms used to evaluate the presence of single-nucleotide polymorphisms (SNPs) in the different target regions.

Supplementary file 2. Primers and crRNA guides used for the SHERLOCK4AAT toolbox.

Supplementary file 3. RPA primers used to produce the in vitro transcribed targets (positive control templates).

Supplementary file 4. Summary of BLAST analysis obtained against the 18S pan-trypanosomatid SHERLOCK amplicon and evaluation of the homology of the target region of each crRNA. The percentage identity of the target amplicon of pan-trypanosomatid RPA was evaluated using different degrees of exclusion, to assess its percentage of identity within the different groups and especially in the Phylum Euglenozoa, excluding the target group Kinetoplastea. The number of sequences evaluated is shown, as well as the percentage of sequences where the target regions of the pan-trypanosomatid guides (crRNAs) are conserved.

MDAR checklist

## 1Data availability

All data generated or analysed during this study are included in the manuscript and supporting files.

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
