## [Editor Report · eLife Assessment]

This **important** study reports an advancement in the diagnosis of Animal African Trypanosomosis (AAT), which adapts a CRISPR-based diagnostic tool (SHERLOCK4AAT) to detect different trypanosome species responsible for AAT. The evidence supporting the conclusions is **convincing** and in line with the current state-of-the-art diagnostics. This study will be of interest to the fields of Epidemiology, Public Health, and Veterinary Medicine.

---

## [Referee Report · Reviewer #1 (Public review)]

Summary:

The authors developed SHERLOCK4AAT, a CRISPR-Cas13a-based diagnostic toolbox for detecting multiple trypanosome species responsible for animal African trypanosomiasis. They created species-specific assays targeting six prevalent parasite species and validated the system using dried blood spots from domestic pigs in Guinea and Côte d'Ivoire. Field testing revealed high infection rates (62.7% of pigs infected) and, notably, the presence of human-infective parasites in domestic animals.

Major Strengths:

This study represents a valuable application of CRISPR-based detection technology to veterinary diagnostics, with strong potential for practical implementation. The authors conducted comprehensive validation, including statistical analyses to determine sensitivity and specificity, and demonstrated field utility through large-scale testing of 424 samples from two geographically distinct regions. The detection of human-infective parasites in pigs at both sites provides important One Health insights supporting integrated disease surveillance and has direct implications for public health policy and disease elimination programs. The methodology is robust, incorporating Bayesian statistical modeling and offering clear practical advantages such as dried blood spot compatibility and detection of active infections. The revised manuscript also addresses implementation considerations, including cost, training needs, and field logistics.

Major Weaknesses:

Some technical limitations constrain broader applicability. The assay for one key parasite species (T. vivax) shows suboptimal sensitivity, which may limit its utility in detecting this important pathogen. The current assay design does not distinguish between closely related species within the same subgenus-an important factor for certain epidemiological studies. Additionally, some assays relied on synthetic controls due to unavailable biological material, and the discussion on potential cross-reactivity with related kinetoplastid parasites is limited.

Achievement of Aims: The authors clearly achieved their primary objectives of developing a sensitive, species-specific diagnostic system and demonstrating its applicability in real-world settings. The detection of human-infective trypanosomes in domestic pigs provides valuable epidemiological evidence in support of One Health strategies and targeted disease elimination efforts.

Impact and Utility:

This work responds to a well-documented need in veterinary diagnostics, where current methods often lack sensitivity or species discrimination. The system offers practical benefits for resource-limited settings through a short assay duration and compatibility with dried blood spot samples. While certain performance limitations may restrict broader adoption, the species identification capability represents a substantial advancement over existing approaches. The findings enhance our understanding of parasite diversity in livestock and their potential role as zoonotic reservoirs, with implications extending beyond veterinary medicine to public health surveillance and policy development.

Context:

This study makes a timely and relevant contribution to diagnostic epidemiology and One Health surveillance frameworks. The field-adapted use of advanced molecular detection technologies represents a significant step toward improved disease monitoring in regions where trypanosomiasis poses ongoing threats to animal health, agriculture, and human livelihoods. The cross-disciplinary implications for veterinary medicine, public health, and disease elimination programs underscore the broader significance of this work.

---

## [Referee Report · Reviewer #2 (Public review)]

Summary:

The manuscript is fundamental due to the significance of its findings. The strength of the evidence is compelling, and the manuscript is publishable since the corrections have been made.

Strengths:

Using a Novel SHERLOCK4AAT toolkit for diagnosis.

Identification of various sub-species of Trypanosomes.

Differentiating the animal sub-species from the human one.

Corrections Made:

Definite articles have been removed from the title.

The words of the title have been reduced to 15.

Typographical errors have been corrected.

Weaknesses:

None

---

## [Referee Report · Reviewer #3 (Public review)]

Summary:

The study adapts CRISPR-based detection toolkit (SHERLOCK assay) using conserved and species-specific targets for the detection of some members of the Trypanosomatidae family of veterinary importance and species-specific assays to differentiate between the six most common animal trypanosomes species responsible for AAT (SHERLOCK4AAT). The assays were able to discriminate between Trypanozoon (T. b. brucei, T. evansi and T. equiperdum), T. congolense (Savanah, Forest Kilifi and Dzanga sangha), T. vivax, T. theileri, T. simiae and T. suis. The design of both broad and species-specific assays was based primarily on sequences of the 18S rRNA, GAPDH (Glyceraldehyde-3-phosphate dehydrogenase) and invariant flagellum antigen (IFX) genes for species identification. Most importantly the authors showed varying limit of detection for the different SHERLOCK assays which is somewhat comparable to PCR-derived molecular techniques currently used for detecting animal trypanosomes even though some of these methodologies have used other primers that target genes such as ITS1 and 7SL sRNA.

The data presented in the study are particularly useful and of significant interest for diagnosis of AAT in affected areas.

Strengths:

The assays convincingly allow for the analysis and detection of most trypanosomes in AAT

Weaknesses:

Inability for the assay to distinguish *T. b. brucei*, *T. evansi* and *T. equiperdum* using the 18S rRNA gene as well as the IFX gene not achieving the sensitivity requirements for detection of *T. vivax*. Both *T. brucei brucei* and *T. vivax* are the most predominant infective species in animals (in addition to *T. congolense*), therefore a reliable assay should be able to convincingly detect these to allow for proper use of diagnostic assay.

---

## [Author Response]

The following is the authors’ response to the original reviews.

**Reviewer #1 (Public review):**
Summary:This study addresses a critical gap in veterinary diagnostics by developing a CRISPR-based diagnostic toolbox (SHERLOCK4AAT) for detecting animal African trypanosomosis. It describes the development and field deployment of SHERLOCK4AAT, a CRISPR-Cas13-based diagnostic toolbox for the eco-epidemiological surveillance of animal African trypanosomosis (AAT) in West Africa.The authors successfully created and validated species-specific assays for multiple trypanosomes, including T. congolense, T. vivax, T. theileri, T. simiae, and T. suis, alongside pan-trypanosomatid and pan-Trypanozoon assays. The field validation in pigs from Guinea and Côte d'Ivoire revealed high trypanosome prevalence (62.7%), frequent co-infections, and importantly identified T. b. gambiense in one animal at each site, suggesting pigs may serve as potential reservoirs for this human-infective parasite.A major strength of the study lies in its methodological innovation. By adapting SHERLOCK to target both conserved and species-discriminating sequences, the authors achieved high sensitivity and specificity in detecting Trypanosoma species. Their use of dried blood spots, validated thresholds through ROC analyses, and statistical robustness (e.g., Bayesian latent class modeling) provides a strong foundation for their conclusions.The results are significant: over 60% of pigs tested positive for at least one trypanosome species, with co-infections observed frequently and T. b. gambiense detected in pigs at both sites. These findings have direct implications for the role of animal reservoirs in human disease transmission and underscore the value of pigs as sentinel hosts in gHAT elimination efforts.The limitations are well acknowledged, particularly the suboptimal sensitivity of the T. vivax assay and the reliance on synthetic controls for T. suis and T. simiae. However, these limitations do not undermine the overall conclusions, and the paper provides a clear roadmap for further assay refinement and implementation.This study offers a timely, impactful, and well-substantiated contribution to the field. The SHERLOCK4AAT toolbox holds promise for improving AAT diagnostics in resource-limited settings and advancing One Health surveillance frameworks.

Thank you

Strengths:(1) The adaptation of SHERLOCK technology for AAT represents a significant technical advancement, offering higher sensitivity than traditional parasitological methods and the ability to detect multiple species simultaneously.(2) Rigorously performed with validation using appropriate controls, ROC curve analyses, and Bayesian latent class modelling, establishing clear analytical sensitivity and specificity for most assays.(3) Testing 424 pig samples across two countries provides robust evidence of the tool's utility and reveals important epidemiological insights about trypanosome diversity and prevalence.(4) The identification of T. b. gambiense in pigs at both sites has significant implications for HAT elimination strategies and highlights the need for integrated One Health approaches.(5) The use of dried blood spots and RNA detection for active infections makes the approach practical for field surveillance in resource-limited settings.

Thank you

Weaknesses:(1) The manuscript would benefit from more detailed discussion of practical considerations such as cost, equipment requirements, and training needs for implementing SHERLOCK in endemic areas and rural settings which would improve applicability.

This is now adressed in the revised discussion (end of the first section).

(2) Limited discussion of pig selection criteria: More justification for choosing pigs as sentinel animals and discussion of potential limitations of this approach would strengthen the manuscript.

Yes, this is now more clearly explained in the revised discussion (beginning of the first section).

(3) More details on why certain genes were targeted would strengthen the methods.

The first result section ‘Selection of targets for broad and species-specific SHERLOCK assays targeting AAT species (SHERLOCK4AAT)’ is already dedicated to extensively explaining target selection, hence we’re afraid we don’t know what could be added.

(4) Table formatting could be improved for readability.(5) Some figures are complex and would benefit from additional explanations in the legends.

We have tried to improve these two aspects as much as possible in the revised manuscript.

**Reviewer #2 (Public review):**
Summary:The manuscript is important due to the significance of the findings. The strength of evidence is convincing.

Thank you

Strengths:(1) Using a Novel SHERLOCK4AAT toolkit for diagnosis.(2) Identification of various sub-species of Trypanosomes.(3) Differentiating the animal subspecies from the human one.

Thank you

Weaknesses:(1) The title is too long, and the use of definite articles should be reduced in the title.

The title has been improved in the revised version.

(2) The route of blood sample collection in the animals should be well defined and explained.

This has been more clearly explained in the revised method section.

**Reviewer #3 (Public review):**
Summary:The study adapts CRISPR-based detection toolkit (SHERLOCK assay) using conserved and species-specific targets for the detection of some members of the Trypanosomatidae family of veterinary importance and species-specific assays to differentiate between the six most common animal trypanosome species responsible for AAT (SHERLOCK4AAT). The assays were able to discriminate between Trypanozoon (T. b. brucei, T. evansi, and T. equiperdum), T. congolense (Savanah, Forest Kilifi, and Dzanga sangha), T. vivax, T. theileri, T. simiae, and T. suis. The design of both broad and species-specific assays was based primarily on sequences of the 18S rRNA, GAPDH (Glyceraldehyde-3-phosphate dehydrogenase), and invariant flagellum antigen (IFX) genes for species identification. Most importantly, the authors showed varying limits of detection for the different SHERLOCK assays, which is somewhat comparable to PCR-derived molecular techniques currently used for detecting animal trypanosomes, even though some of these methodologies have used other primers that target genes such as ITS1 and 7SL sRNA.The data presented in the study are particularly useful and of significant interest for the diagnosis of AAT in affected areas.

Thank you

Strengths:The assays convincingly allow for the analysis and detection of most trypanosomes in AAT.

Thank you

Weaknesses:Inability for the assay to distinguish *T. b. brucei*, *T. evansi*, and *T. equiperdum* using the 18S rRNA gene, as well as the IFX gene, not achieving the sensitivity requirements for detection of *T. vivax*. Both *T. brucei brucei* and *T. vivax* are the most predominant infective species in animals (in addition to *T. congolense*), therefore, a reliable assay should be able to convincingly detect these to allow for proper use of the diagnostic assay.

We agree with this point and aim to improve the toolbox for future studies.

**Reviewer #1 (Recommendations for the authors):**
(1) Provide additional details on the practicality of SHERLOCK deployment in the field, including training, costs, and infrastructure (potential challenges for field deployment, including suggestions for how to overcome these barriers).

This is now adressed in the revised discussion (end of the first section).

(2) Provide more detailed justification for choosing pigs as the main study species and discuss potential benefits and limitations of extending the approach to other livestock species.

Yes, this is now more clearly explained in the revised discussion (beginning of the first section).

(3) Add a comparison table comparing SHERLOCK4AAT performance metrics (sensitivity, specificity, LoD) with existing molecular diagnostic methods for AAT for ease of reference.

There are dozens of different serological, immunological and molecular approaches with highlty variable levels of sensitivity and specificities already reviewed and compared in detail in two references from 2022 (Desquesnes et al. a and b), which we have cited, as well as in a newly added reference (EBHODAGHE F acta trop 2018). Hence, we decided to only refer to the most comparable studies in the present article.

(4) Review complex figures and improve legends for better readability and interpretation.

We have tried to improve this as much as possible in the revised manuscript.

**Reviewer #2 (Recommendations for the authors):**
(1) Reduce the number of words in the title from 28 to not more than 20.

The title has been improved in the revised version.

(2) Specify the particular route of collection of blood samples in the various animals.

Yes, this is now more clearly explained in the revised method section.

(3) Correct all typographical errors.

We have tried to improve this as much as possible in the revised manuscript.

Thanks. I wish you the best in your publication process.

Thank you

**Reviewer #3 (Recommendations for the authors):**
Minor comments(1) The authors can expand the discussion to include other recent diagnostic assays for Animal trypanosomiasis, such as those that target other genes like tubulin.Please see response to Review 1 point #3 above.(2) The cost-effectiveness of the use of the assay can be discussed since the assay is expected to be used for work in some resource-deprived areas. For example, will it cost a researcher less to do a diagnosis with this assay relative to what is already available?

This is now adressed in the revised discussion (end of the first section).

(3) Is Cote d'Ivoire more endemic for AAT than Guinea? Will this account for the apparently consistent differences in the percentage of positive samples, or just because of the type of samples used from the two locations?

As the sampling method, sample preservation and sample analysis were the same for both groups - yes, it appears that pigs, at least for domesticated ones, in the study region of Cote d'Ivoire were more frequently infected than those in the study region of Guinea. It is however risky to extrapolate these observations to the AAT prevalence in the entire countries and/or to other mammals.

(4) Can the authors comment on how long one can store the samples for an effective and reliable assay?

The samples can be stored for several months at ambient temperature in a sealed bag with silica gel packages to reduce humidity. We have added this detail in the revised methods section.

(5) It is not clear whether the authors used conventional molecular diagnostics to compare the data obtained from this particular cohort of animals as reference is made to published data. It is not surprising that the SHERLOCK performed better than using parasitology-based methodology.

This is now adressed in the revised discussion.

(6) (Figure 4D-5D) should be 4D and 5D.

Thank you, this has been corrected.